# MC-UNet: Martian Crater Segmentation at Semantic and Instance Levels Using U-Net-Based Convolutional Neural Network

**Dong Chen [1], Fan Hu [1], P. Takis Mathiopoulos [2], Zhenxin Zhang [3,\*] and Jiju Peethambaran [4]**

1   College of Civil Engineering, Nanjing Forestry University, Nanjing 210037, China
2   Department of Informatics and Telecommunications, National and Kapodistrian University of Athens, 15784 Athens, Greece
3   College of Resource Environment and Tourism, Capital Normal University, Beijing 100048, China
4   Department of Mathematics and Computing Science, Saint Mary's University, Halifax, NS B3P 2M6, Canada
\*   Correspondence: zhangzhx@cnu.edu.cn; Tel.: +86-13439045960

**Abstract:** Crater recognition on Mars is of paramount importance for many space science applications, such as accurate planetary surface age dating and geological mapping. Such recognition is achieved by means of various image-processing techniques employing traditional CNNs (convolutional neural networks), which typically suffer from slow convergence and relatively low accuracy. In this paper, we propose a novel CNN, referred to as MC-UNet (Martian Crater U-Net), wherein classical U-Net is employed as the backbone for accurate identification of Martian craters at semantic and instance levels from thermal-emission-imaging-system (THEMIS) daytime infrared images. Compared with classical U-Net, the depth of the layers of MC-UNet is expanded to six, while the maximum number of channels is decreased to one-fourth, thereby making the proposed CNN-based architecture computationally efficient while maintaining a high recognition rate of impact craters on Mars. For enhancing the operation of MC-UNet, we adopt average pooling and embed channel attention into the skip-connection process between the encoder and decoder layers at the same network depth so that large-sized Martian craters can be more accurately recognized. The proposed MC-UNet is adequately trained using 2∼32 km radii Martian craters from THEMIS daytime infrared annotated images. For the predicted Martian crater rim pixels, template matching is subsequently used to recognize Martian craters at the instance level. The experimental results indicate that MC-UNet has the potential to recognize Martian craters with a maximum radius of 31.28 km (136 pixels) with a recall of 0.7916 and $F_1$-score of 0.8355. The promising performance shows that the proposed MC-UNet is on par with or even better than other classical CNN architectures, such as U-Net and Crater U-Net.

**Keywords:** Martian craters; crater recognition; semantic segmentation; instance-level segmentation; U-Net; template matching





## 1. Introduction

Impact craters exist on all the terrestrial planets, i.e., Mercury, Venus, Earth and Mars, and they constitute a core part of the widespread geomorphology in the solar system. Figure 1 illustrates an image of such impact craters that exist on Mars obtained through the THEMIS thermal infrared system. Data obtained from observing such craters provide extremely valuable information for astronomical research, including the formation and evolution of celestial bodies [1], rotation axis changes [2], magmatic activity [3] and past catastrophic events [4]. Impact-crater statistics, such as their number, density, size–frequency distribution, topographic properties and interior and ejecta morphologies, are vital data from which we can infer the relative geological age [5] of different surface regions and analyze the change process of the planetary surface. For example, "crater counting" is a technique planetary scientists typically use to establish the relative ages of planetary surfaces.

This method first estimates craters' size–frequency relationship by calculating the number and size of impact craters on the planet's surface. Assuming that the number of craters and the averaged crater production rate, which is considered to be constant throughout the geological history, are known, then the "surface age" can be inferred [6]. If the crater age has been determined, we have the capability to establish links between crater ages with some hot research nowadays. For instance, we can locate the possible source crater (ejection location) of Martian meteorites, which provides fundamental insight into the interior composition, evolution and geological characteristics of different locations on Mars [7]. Ancient Martian meteorites such as ALHA84001 include evidence of aqueous alteration [8]. The current evidence of fresh craters reveals current-day Martian meteorites [9].

Furthermore, by analyzing the geometric shape, taper and impact metamorphism of the craters, the impact direction, velocity and mass of the meteorite before the collision can be estimated. Accurate crater data can also be used for planetary mapping and landmark-based autonomous navigation to identify accurate landing sites and map their surrounding topography for planning outer space exploration activities, such as circling, landing, lifting off and flying over study sites.

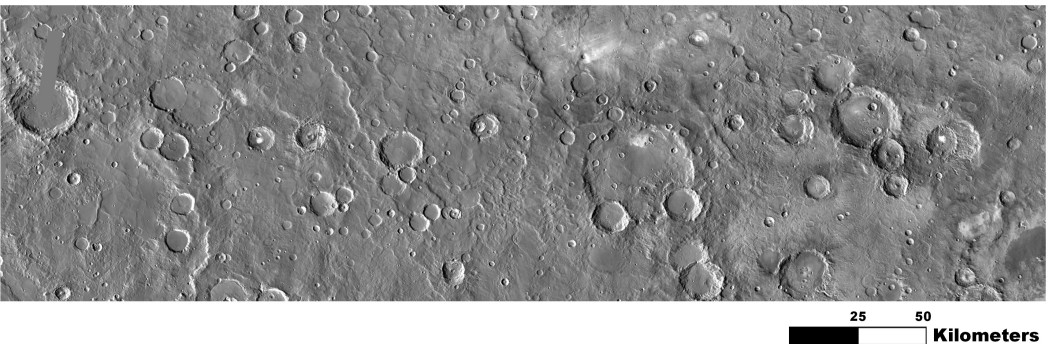

**Figure 1.** Impact craters on Mars in THEMIS thermal infrared images.

A problem of paramount importance for current planetary exploration research activities, such as topographical representation, geomorphological analysis, preservation state estimation as well as relative and/or absolute surface-age dating, is the accurate and efficient detection of craters from planet-surface images obtained via remote-sensing techniques. Traditional crater segmentation methods are based on rather classical computer-vision algorithms, such as the Canny edge contouring algorithm [10,11], Hough transform (HT) [12–14] and template matching [15,16]. For instance, Cheng et al. [11] used the Canny operator with two types of kernels to detect fuzzy and sharp rims for different sizes of craters. Sawabe et al. [13,14] employed fuzzy HT to extract discrete or broken circular topographic features from lunar crater images. It has been demonstrated that fuzzy HT detects craters of all sizes and shapes with 80% accuracy without parameter tuning. Kim et al. [15] presented another approach by combining edge detection, template matching and a neural-network-based scheme to identify the impact craters on Mars. Viking Orbiter (VO), Mars Orbiter Camera (MOC) and High-Resolution Stereo Camera (HRSC) images have shown that a recognition rate of 70~90% can be obtained. It is underlined that, although each step of the above-mentioned methods can be well-explained and visualized, and the obtained performance evaluation results have shown promising crater detection ability for Mars and/or Moon images, their frameworks are complex because a plethora of steps are loosely coupled in the method pipeline. In addition, many of these methods suffer from variations of crater appearances, which are typically caused by illumination and incidence angle, bright-to-dark shading pattern, crater geomorphological shape, crater surrounding topography and atmospheric state.

## 1.1. U-Net for Image Segmentation

Nowadays, there exist numerous deep-neural-network-based segmentation techniques, including semantic, instance- and part-level segmentation. As the basis of fine-grained level segmentation, e.g., instance- and part-level segments, coarse-grained semantic segmentation is of paramount importance through solving a series of complex computer-vision tasks. For example, the model U-Net [17]—designed for medical image semantic segmentation and whose full-layer skip connections supplement the loss of details caused by downsamplings—has the advantages of elegant architecture, high predictive accuracy, strong scalability, high training efficiency as well as lesser training size requirements and model parameters. U-Net performs well in connection with binary segmentation tasks as these have relatively simple semantic context and emphasize accurate boundary delineation. Because of these characteristics, U-Net and its variants have achieved remarkable segmentation results in medical image segmentation [18], coastline mapping [19] and urban building segmentation for Digital Twin applications [20]. In particular, Saeedizadeh et al. in [18] proposed a variant of U-Net, referred to as VT-UNet, that detects infected chest regions through adding total-variation as a regular term into the loss function. The proposed method suppresses the discontinuous regions and makes the infected regions form connected components, thereby improving the mIoU to 0.994. Heidler et al. in [19] proposed a new model, called HED-UNet, which combines the classical U-Net with a holistically nested edge detection (HED) architecture [21] for simultaneous segmentation and edge delineation of Antarctic coastline. Finally, Yu et al. in [20] introduced an improved attention-gate mechanism into the U-Net architecture, thus achieving building segmentation from corresponding high-resolution images.

## 1.2. U-Net for Crater Segmentation

Since planetary crater segmentation is also a binary segmentation task, the design focus should be on maintaining the precise boundary between foreground rim edges and non-rim background. Therefore, many CNNs, including U-Net [17] and its variants, have been widely used in crater segmentation and recognition tasks. For example, Silburt et al. in [22] employed the U-Net model with three-times downsamplings for the improvement of crater semantic segmentation. Furthermore, a template matching algorithm [23] has been utilized to automatically identify the impact craters. Performance evaluation results have shown that this method was able to correctly identify 92% of craters in the dataset of the lunar digital elevation model (DEM) [24]. The work reported in [22] is very important as it is the first research effort to introduce deep learning techniques into the semantic segmentation of craters, and it also established the crater segmentation paradigm through "U-Net + template matching". Later on, DeLatte et al. in [25] devised a new model, termed Crater U-Net, by increasing the number of downsampling times to seven and introducing dropout regularization into the U-Net model to obtain a lightweight segmentation network for crater semantic segmentation, followed by individual crater recognition using template matching. This method achieved a matching accuracy of 65~76% depending on the volume of the training image titles involved. Inspired by the U-Net++ [26] architecture, which is based on nested and dense skip connections as well as Attention U-Net [27], which is based on an attentional gating mechanism, Jia et al. in [28] presented NAU-Net, whose recognition rate of craters in lunar DEM images is about 3% higher than that of the classical U-Net model. Finally, Hong et al. in [29] further improved the crater recognition rate by 6% as compared to classical U-Net by employing data augmentation, i.e., constructing multilayer pyramid images from the lunar DEM dataset.

## 1.3. Unresolved Issues

Although the above-mentioned methods have enhanced U-Net variants to achieve good results for crater segmentation and recognition tasks, some outstanding issues and limitations still exist. In particular, for their structure setting, these works adopt a rather ad hoc approach with different frenquencies of downsampling and a varying number of

channels. Thus, the two questions that naturally arises are: (i) What is the optimal size of the receptive field that can more accurately summarize crater rim semantics? and (ii) What is the maximum number of framework channels that has sufficiently discriminative ability to accurately infer crater rims while maintaining the lightweight characteristics of the framework? As far as the mechanism setting is concerned, there are still some improvements to be made. For example, although dropout regularization in Crater U-Net [25] reduces the number of model parameters, it makes it difficult for model training to converge and, therefore, needs more training epochs. Although NAU-Net [28] combines the core mechanisms from U-Net++ [26] and Attention U-Net [27] to achieve a higher recognition rate than classical U-Net, the model tends to be cumbersome and has an excessive number of parameters. Thus, an effective framework for crater segmentation with a concise structure, a faster convergence speed and a high recognition rate are needed.

### 1.4. Novel Contributions

By considering the Martian crater segmentation problem, we employ classical U-Net as the backbone to devise a new encoder–decoder architecture from the perspective of structure and mechanism. Its main novelty is that the proposed Martian Crater U-Net (MC-UNet) is tailored to the task of Martian crater segmentation at both semantic and instance levels. We analyze and evaluate the effects of the network depth and the number of convolution layer channels on segmentation accuracy to determine the optimal architecture structure. On this basis, for MC-UNet's mechanism design, we fully consider different downsampling strategies, encoding and decoding feature-fusion methods and attention mechanisms to explore an optimal U-Net operational framework. In the context of the proposed MC-UNet CNN operational framework, the main novel contributions of our paper can be summarized as follows:

(1) We present a novel convolutional neural network, i.e., MC-UNet, that is tailored to the task of crater recognition from Mars THEMIS images. The created model MC-UNet has high recognition accuracy of craters, especially when recognizing large-sized craters. Further, MC-UNet is lightweight with fewer parameters, thereby achieving a fast convergence rate with fewer epochs.

(2) We propose a novel, hierarchical-based approach to create the proposed MC-UNet CNN. In particular, first, the structure of the encoder–decoder architecture of MC-UNet is devised, followed by integration of the most-appropriate mechanisms, including downsampling, feature-map fusion and attention. This hierarchical strategy is a particularly effective way to construct an optimal neural network for the task of Martian crater recognition from THEMIS images.

(3) We adopt template matching for the implementation of instance-level Martian crater recognition. By imposing the template matching operation on the prediction map that is predicted by the proposed MC-UNet, potential craters can be progressively identified according to their size.

This paper is organized as follows: Section 2 describes the detailed methodology, including semantic segmentation of impact crater rims using the architecture of MC-UNet and instance segmentation of impact craters by the technique of template matching. In Section 3, the experimental dataset and the performance evaluation results of the detected impact craters from THEMIS daytime infrared mosaics are presented, analyzed and discussed. Finally, Section 5 concludes the paper while providing a few suggestions for future research topics.

## 2. Methodology

### 2.1. MC-UNet Framework

For Martian crater segmentation, we propose a novel semantic convolutional neural network, i.e., MC-UNet, which uses classic U-Net as a backbone. As shown in Figure 2, cropped tiles with $512 \times 512 \times 1$ THEMIS images are fed to MC-UNet as inputs. In each encoding layer, the inputs are downsampled, and input features are progressively increased

using consecutive convolutional operations and average pooling. After 5 rounds of autoencoding, the size of inputs shrinks to $8 \times 8$ and the channel is expanded to 256. In each decoding layer, we use a $3 \times 3$ transposed convolution to align the sampled feature map of the corresponding layer from the encoder. The local features generated by the encoder are concatenated with the corresponding upsampling features through skip connection and an attention mechanism. MC-UNet outputs a feature map with a size of $512 \times 512 \times 1$, from which each pixel represents the possibility of it being a Martian crater contour.

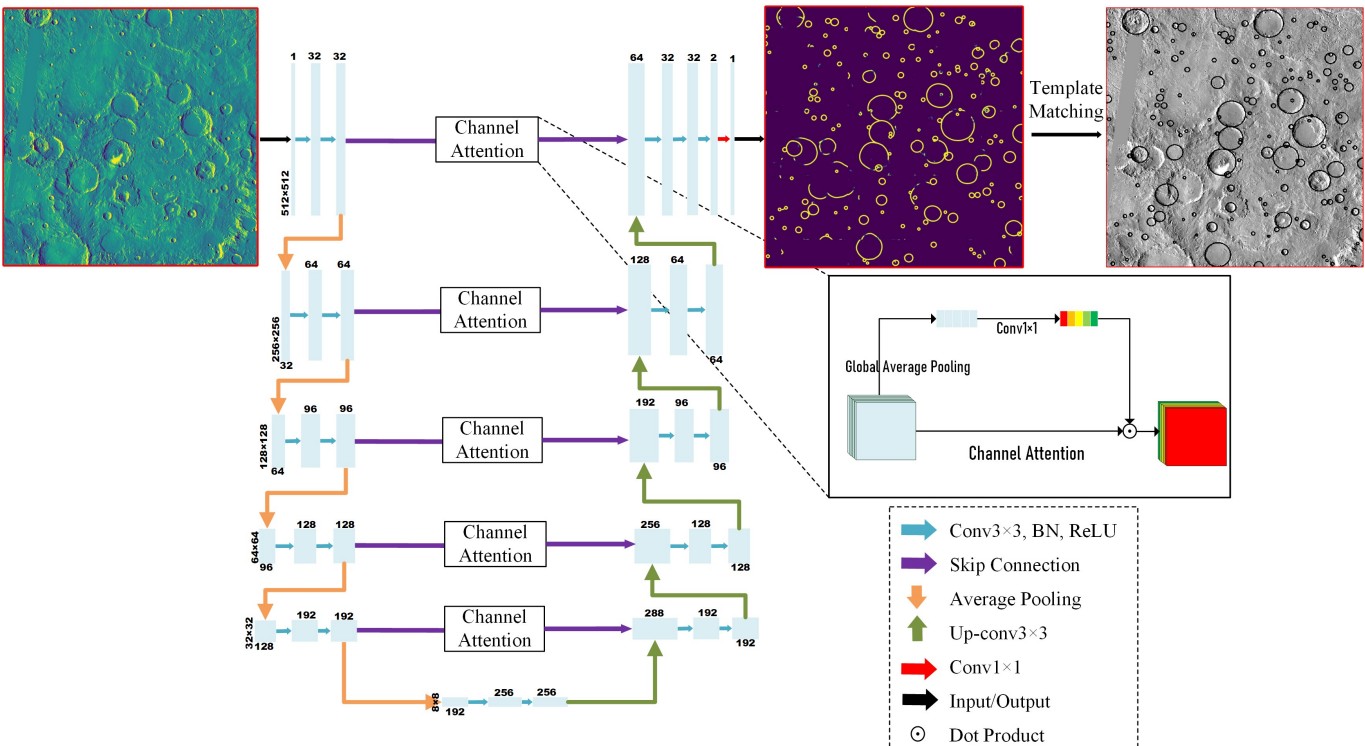

**Figure 2.** Framework of the proposed MC-UNet.

### 2.2. Structure of MC-UNet

Typically, the structure of a CNN is roughly described in terms of its "depth" and "width", which are of paramount importance for accurate recognition of Martian craters. Generally, the depth is defined as the number of the network's layers, while the width of a network is defined as the maximum number of nodes/neurons in a layer. However, in this paper, we redefine these two terms to avoid ambiguity. The term 'depth' of a network refers to the number of times downsampling, while the 'width' is the maximum number of channels of a layer. As the depth affects the sizes of receptive fields, choosing a reasonable depth can guarantee the neural network has an adequate receptive field size, which contributes to the improvement of semantic segmentation accuracy. The width affects the generalization ability of model fitting. Having a large or small number of channels (feature maps) tends to cause the problems of over- or under-fitting, respectively. In practice, the appropriate values for depth and width are acquired mostly based on trial-and-error experiments. We determine the optimal MC-UNet structure based on a hierarchical training strategy. That is, we first determine the best depth and fix this depth to infer the optimal width. The structure of MC-UNet can be represented as follows:

$$\mathcal{P} = \mathcal{M}_D^{[d]}[\mathcal{M}_E^{[d]}(\mathcal{X}_{<W,H,\omega \times C>})] \tag{1}$$

where the parameters $\mathcal{M}_D$ and $\mathcal{M}_E$ represent the encoder and decoder modules at each layer, respectively. The depth of the layer is represented by the parameter $d$. The parameter $\mathcal{X}$ is the input feature map, whose width, height and channel number are represented by

the parameters *W*, *H* and *C*, respectively. The parameter $\omega$ is defined as an expansion coefficient and determines how many times the width of the neural network increases. In this paper, we take a tensor $\mathcal{X}$ with dimensions of $<512, 512, 1>$ as input, and we output a probability map $\mathcal{P}$ with the same size of $\mathcal{X}$ after five repeated encoding and decoding procedures. The encoding and decoding modules can be defined by

$$\mathcal{M}_E = \mathcal{F}_{pooling} \otimes \mathcal{F}_{conv} \otimes \mathcal{F}_{conv}(\mathcal{X}_{<W,H,\omega \times C>}) \tag{2}$$

$$\mathcal{M}_D = \mathcal{F}_{conv} \otimes \mathcal{F}_{conv} \otimes \mathcal{F}_{concate} \otimes \mathcal{F}_{upconv}(\mathcal{X}_{<W,H,\omega \times C>}) \tag{3}$$

where $\mathcal{F}_{pooling}$ represents average pooling, $\mathcal{F}_{conv}$ represents a convolution operation using a $3 \times 3$ kernel, $\mathcal{F}_{concate}$ represents the concatenation of feature maps from an encoder–decoder architecture at the same layer, $\mathcal{F}_{upconv}$ represents a transposed convolution with a $3 \times 3$ kernel, and the symbol "$\otimes$" represents an assembly operation that joins more components to form a complete encoder or decoder module. To optimize the structure of Equation (1), it needs to simultaneously tune parameters of $d$, $\omega$, *W* and *H* [30]. In practice, we only tune the parameters $d$ and $\omega$ in parallel and intentionally ignore the other two parameters *W* and *H* because the change of input $\mathcal{X}$'s resolution *W* and *H* is beyond the scope of the neural network's structural optimization. To optimize the network's structure, we construct six types of candidate models with structure depths varying from 2 to 7. For each candidate model, its width is set to the range from 16 to 128, which is a value typically adopted by Crater U-Net [25]. The candidate model with the best prediction performance in terms of segmentation accuracy, model size and training efficiency is selected, and at this time, the corresponding depth $d^*$ is supposed to be optimal. Subsequently, we fix the selected depth $d^*$ to expand the width by multiplying three expansion coefficients $\omega$, e.g., 1, 1.5 and 2. The width of the model that shows the best prediction performance is chosen as the optimal value ($\omega^*$). In summary, the above structural-optimization procedure using a hierarchical strategy tries to maximize the equation expressed below:

$$d^*, \omega^* = \arg\max_{d,\omega}(acc(\mathcal{P})) \tag{4}$$

$$s.t. \quad d \in \{2,3,4,5,6,7\}; \ \omega \in \{1,1.5,2\}$$

where $d^*$ and $\omega^*$ denote the optimal depth and expansion coefficient, respectively. The symbol $acc(\cdot)$ represents the model's performance for the task of Martian crater segmentation at both semantic and instance levels.

### 2.3. Mechanism of MC-UNet

It is extremely difficult to significantly improve crater segmentation accuracy by merely tuning the structure of a convolutional neural network [30]. To solve this problem, inspired by the works of ResNet [31] and DenseNet [32], some researchers have modified the convolutional modules of U-Net with a residual learning form and densely connected style for realizing the construction of Res-UNet [33] and Dense-UNet [34], respectively. Similarly, Attention U-Net [27] introduces the idea of an attention gate in the skip connection, which improves the prediction performance of U-Net on two large CT abdominal datasets. In the task of planetary crater segmentation, Jia et al. [28] proposed a novel NAU-Net that combines U-Net++ [26] and an attention mechanism to identify impact craters. The created model can recognize unusual and complex craters, such as smaller, larger and/or overlapping ones. Further, comparative analysis with U-Net [17], UNet++[26], Dense-UNet [34], Attention U-Net [27] and R2-UNet [35] shows that these improved U-Net architectures are inferior to NAU-Net [28] because these U-Net variants are designed for and trained on different types of datasets rather than on impact crater images, making these models' generalization ability inferior and unreliable for the segmentation of impact craters. For the task of impact-crater recognition, we adopt three types of mechanisms into MC-UNet,

namely, a downsampling strategy, feature map fusion, and an attention mechanism to achieve better recognition of Martian craters.

(1) Downsampling strategy: Downsampling is indispensable in semantic segmentation networks and is usually behind the convolutional layers. Choosing an appropriate downsampling strategy can help the deep learning network expand the receptive field to obtain semantic information at a high level, thereby enhancing the ability of information representation. Some classic semantic segmentation models, e.g., FCN [36], U-Net [17] and SegNet [37], adopt max-pooling downsampling, which selects the maximum value in the receptive field of the pooling kernel. Max pooling emphasizes the foreground information more, i.e., impact crater rim contours of the feature maps; therefore, it strengthens the ability of feature encoding and aggregation of the convolutional layers and enhances the model's invariance to rotation, translation and scaling. In contrast, average pooling takes the average of all values in the receptive field of the pooling kernel, which implies that the background non-Martian crater contours are more likely to be retained. Although downsampling has strong capability in feature aggregation, one should be aware that the details of the spatial information tend to be lost during a pooling operation. To address such situations, after the traditional convolutional operation, PeleeNet [38] adopts a stem-block module that concatenates the results of max pooling and two consecutive convolutions to finalize the high-level encoding and decrease the loss of information. In this paper, we evaluate Martian crater prediction performance of three pooling strategies, i.e., max pooling, average pooling and stem-block downsampling, from which average pooling is identified as the best one and is hence integrated into MC-UNet for the task of impact crater recognition. We recommend the readers to follow the experiments in Section 3.5 for details.

(2) Feature-map fusion mechanism: The proposed MC-UNet adopts U-Net as a backbone, and it uses a classic skip architecture to combine feature maps from different non-sequential layers. It combines semantic information from the deep, coarse layers and appearance information from the shallow, fine layers [36]. Shallow feature maps tend to capture the overall coarse information, e.g., location, shape and region of impact craters, while more complex patterns such as crater topographic properties and interior morphology are possibly discovered in deeper layers. Figure 3 shows two styles of feature-map fusion, i.e., element-wise adding (Figure 3a) and channel-wise concatenation (Figure 3b). The former is the addition of feature maps and retains the number of channels. The latter is an effective way to stack different feature maps together, which increases the number of channels. These two fusions have been extensively adopted in some classic works. For example, U-Net [17] uses a channel concatenation fusion strategy, while element-wise addition is frequently used in residual networks [33], which are created by stacking multiple residual blocks together to avoid model degeneration with increasing depth. In addition, Crater U-Net [25] also employs element-wise fusion for automatic crater detection on Mars. Inspired by U-Net [17], MC-UNet fuses shallow and deep feature maps by concatenation in the skip-connection architecture, followed by two consecutive convolution operations to merge these feature maps. Through experiments in Section 3.5, we observe that the concatenation fusion style embedded in MC-UNet demonstrates high prediction accuracy not only on semantic crater segmentation but on instance-level crater identification.

(3) Attention mechanism: An attention mechanism uses the attention map to weight the feature maps to obtain more refined and highlighted output. More specifically, the attention map is multiplied element-wisely with the input feature map to highlight the regions of interest of input images. Generally, the commonly used attention mechanisms are roughly classified into channel attention, spatial attention and hybrid attention, which integrates channel and spatial attentions. The above three attentions are exhibited in Figure 4. Channel attention (Figure 4a) uses global average pooling to squeeze the spatial information of each layer into one value. Afterward, it commonly uses $1 \times 1$ convolution to learn a nonlinear and non-mutually exclusive

relationship between channels. Spatial attention (Figure 4b) uses average pooling or max pooling operations along the channel axis to squeeze the multiple feature maps into a new spatial context. Then, it usually uses $1 \times 1$ convolution operations multiple times to learn attention weights, which are multiplied by the input to highlight the informative part of each feature map. Hybrid attention, such as the Convolutional Block Attention Module (CBAM) [39], combines both channel attention and spatial attention in a cascading way to strengthen the extracted feature maps, which provides complementary clues in both channel and spatial dimensions (Figure 4c). In the proposed MC-UNet, we embed these three types of attentions into the skip mechanism of MC-UNet. The experiments show that channel attention can compensate for the loss of fine details in downsampling layers to some extent and exhibits better recognition performance, especially for large-sized Martian craters.

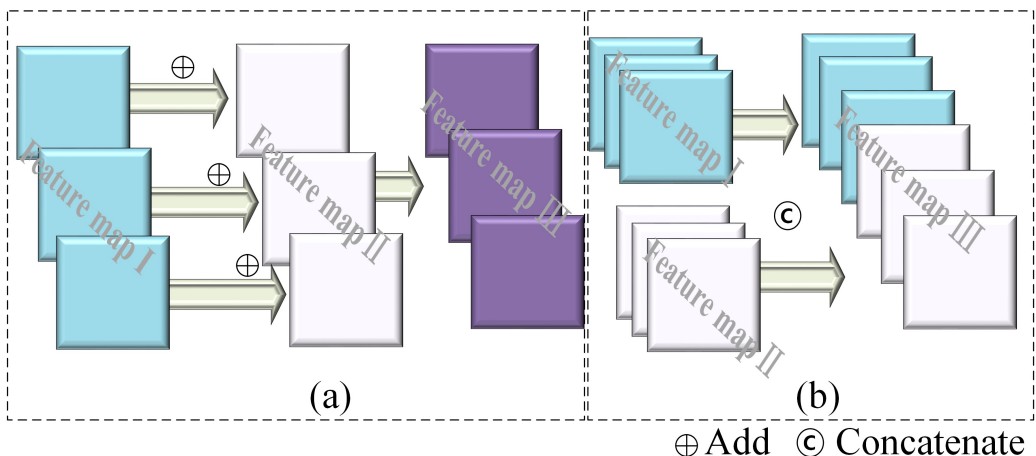

**Figure 3.** Two types of feature-map fusion mechanisms: (**a**) element-wise addition and (**b**) concatenate channel fusion.

### 2.4. Template Matching

As the semantic segmentation result is predicted at pixel level, it is impossible for it to provide instance-level statistics, such as crater position, crater diameter, and crater size-frequency distributions. To solve this problem, we adopt template matching for instance-level crater recognition and further implement the accuracy evaluation based on each identified crater entity. More specifically, we first convert the semantic probability map derived by MC-UNet into a binary map $\mathcal{B}$ with a threshold of 0.4. A circular template $\mathcal{T}$ with a specific size slides through the binary map to identify the same-sized impact craters. Then, we repeat the process by progressively increasing the size of the circular template. All of the Martian craters can be detected in ascending order by their size. The identified crater candidates are further refined by removing the redundant craters according to similarity-based analysis. The detailed matching process is described below:

(a) Circular template generation: We manually create a matrix with size $2(\mathcal{R}_i + d/2) \times (\mathcal{R}_i + d/2)$ ($\mathcal{R}_i \in [r_{min}, r_{max}]$, and $d$ is the thickness of the ring) as a base plate. Based on this plate, we draw a circle centered on the plate with a radius of $\mathcal{R}_i$. The circle's thickness (the difference between the radii of inner and outer enclosing circles) is set to $d/2$, and the corresponding pixel values are set to "1", while others are set to "0". In this way, we obtain a circular template $\mathcal{T}$ with a radius of $R_i$ and thickness of $d/2$. Considering that the ground truth includes annotations of 2 to 32 km radius Martian craters, we set the range of the circle template as $r_{min} = 9$ pixels and $r_{max} = 139$ pixels.

(b)   Circular template matching: The circular template $\mathcal{T}$ slides across the whole binary map $\mathcal{B}$. The normalized cross-correlation coefficient $\mu$ between the covered binary map $\mathcal{B}$ and the circular template $\mathcal{T}$ is presented in Equation (5):

$$\mu(x,y) = \frac{\sum_{i,j=1}^{n}\{\mathcal{T}(i,j)\cdot\mathcal{B}(x+i,y+j)\}}{\sqrt{\sum_{i,j=1}^{n}\mathcal{T}(i,j)^2 \cdot \sum_{i,j=1}^{n}\mathcal{B}(x+i,y+j)^2}} \tag{5}$$

where $\mathcal{T}(i,j)$ and $\mathcal{B}(x,y)$ represent the circular template and the binary map, respectively. The elements from two tuples $(i,j)$ and $(x,y)$ represent the coordinate values of $\mathcal{T}$ and $\mathcal{B}$ on the horizontal and vertical axes, respectively. It should be noted that $\mathcal{T}(i,j)\in[0,1]$ and $\mathcal{B}(x,y)\in[0,1]$. If the coefficient $\mu(x,y)$ is greater than the pre-defined threshold $\mathcal{T}_{\mu}$ ($\mathcal{T}_{\mu}=0.5$), the matching is successful. At this time, statistics such as the template's radius and position are recorded. The procedures from (a) to (b) are repeated until all the Martian craters with a radius from $r_{min}$ to $r_{max}$ have been detected.

(c)   Martian crater candidate refinement: The detected crater candidates with varying radii should be further refined by removing the redundant craters according to similarity-based analysis. More specifically, all the detected Martian crater candidates undergo a pairwise comparison to verify whether redundant Martian craters exist. That is, we compare the consistency of position and radius of a pair of craters. We can draw a safe conclusion that one of the impact craters is redundant if they satisfy the the conditions below

$$\|\mathcal{C}_i - \mathcal{C}_j\|^2 < \mathcal{T}_p \wedge \|\mathcal{R}_i - \mathcal{R}_j\|^2 < \mathcal{T}_r \tag{6}$$

where $\mathcal{C}_i$ and $\mathcal{C}_j$ represent the coordinates of the centers of two Martian crater candidates and $\mathcal{R}_i$ and $\mathcal{R}_j$ represent the radii of crater candidates $i$ and $j$, respectively. $\mathcal{T}_p$ and $\mathcal{T}_r$ are two predefined thresholds. In our case, we set $\mathcal{T}_p = 1.8$ pixels and $\mathcal{T}_r = 1.0$ pixels. If the conditions in Equation (6) are true, we just retain the crater candidate with the highest normalized cross-correlation coefficient calculated by Equation (5).

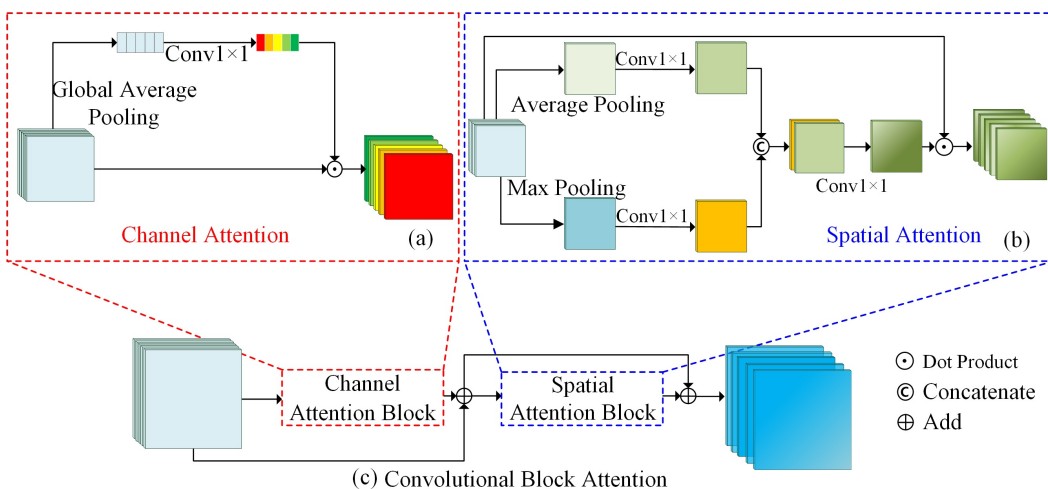

**Figure 4.** Attention mechanisms: (**a**) channel attention, (**b**) spatial attention and (**c**) hybrid attention CBAM [39].

## 3. Performance Evaluation

### 3.1. Dataset Description

In this paper, we select two time periods of Martian THEMIS images released by Mars Space Flight Facility on 16 November 2006 (THEMIS 2006) and 23 June 2010 (THEMIS 2010) as experimental datasets. THEMIS is an instrument equipped on the NASA's Mars

Odyssey spacecraft, and it acquires both visible and thermal daytime infrared images of the Martian surface. These two datasets are processed by a series of processes, i.e., matching, stitching, smoothing and manual annotation, before being released. THEMIS 2006 and 2010 are created using the same Mercator cylindrical projection but with different resolutions, i.e., 230 m and 99 m, respectively. The size of global THEMIS 2006 mosaic images (see Figure 5a) is 92,160 × 46,080 pixels, from which the areas within 30 degrees in north and south latitudes of the equator are separated into 24 square tiles of the same size, each of them spanning 30° in both longitude and latitude. The resolution of each tile is 7680 × 7680 pixels. These 24 individual tiles are commonly regarded as the key research areas for detection of Martian craters due to a low degree of missing data. Another reason is that the impact craters distributed within ± 30° latitudes remain circular in cylindrical projection; however, in high latitude areas, circular craters may be stretched to elliptical shapes due to projection distortion.

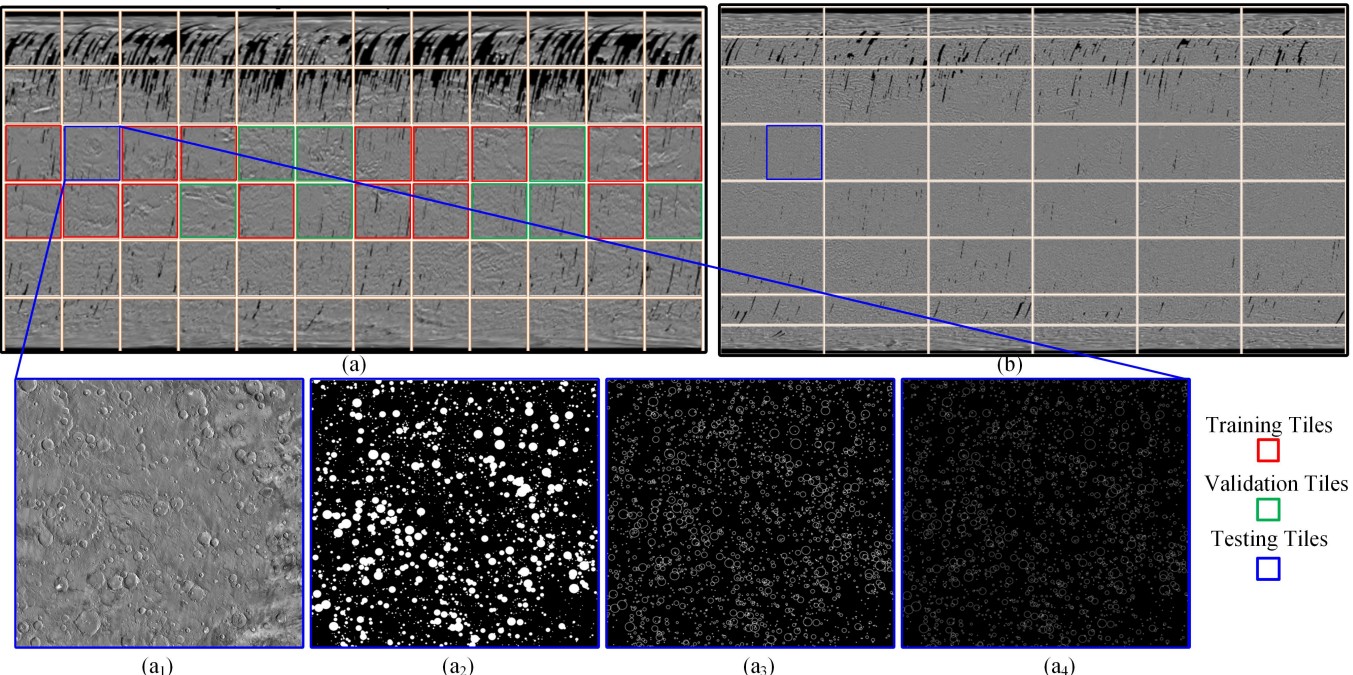

**Figure 5.** Two time periods of THEMIS image datasets. The global THEMIS mosaic images released on 16 November 2006 and 23 June 2010 are shown in subfigures (**a**) and (**b**), respectively. The enlarged views from (**a₁–a₄**) are the zoomed test tile, annotated circles in the solid style, annotated circles with thickness of 8 pixels, and annotated circles with thickness of 4 pixels, respectively.

For the global THEMIS 2010 mosaic images in Figure 5b, the same areas between 30 degrees north and south latitude toward the equator are divided into 12 individual tiles of the same size, from which each individual tile spans 60° of longitude and 30° of latitude. Considering the resolution of THEMIS 2010 is 99 m, the size of each individual tile is 35,565 × 17,783 pixels.

We use the largest and most complete catalog database of Martian craters, i.e., RH2012 [40], as the annotated labels. The catalog records impact crater statistics of 384,343 craters with diameters greater than 1 km. For each crater, the catalog records latitude, longitude, diameter and other relevant information if it can be determined. Based on this catalog, Delatte et al. [25] annotated craters with radii from 2 to 32 km from 24 tiles of THEMIS 2006. That is, the impact craters were manually labeled using circles with different thickness, as evident in the enlarged views in Figure 5. In our case, we select circular annotations with thickness of 8 pixels for training the proposed MC-UNet. More specifically, in the 24 annotated tiles from the THEMIS 2006 dataset, we randomly select 15 tiles as training data, 8 tiles as validation data, and the last one as a test datum. These

tiles are indicated by red, green and blue rectangles in Figure 5a, respectively. For THEMIS 2010, we do not use it for training due to the lack of annotations but instead it is used as test data to verify the robustness of the proposed MC-UNet (see Section 3.9).

### 3.2. Evaluation Metrics

The output of MC-UNet is a probability map from which the value of each pixel represents the probability of each pixel being a Martian crater contour pixel. The values of the probability map are between 0 and 1. Based on the values, we binarize the probability map into a binary map that includes two types of semantic classes, i.e., crater contours and non-crater contours, with a threshold of 0.5. After that, we comprehensively evaluate the overall accuracy of the binary map at the pixel level according to Equation (7):

$$Accuracy = \frac{TP + TN}{TP + TN + FP + FN} \tag{7}$$

where $TP$, $TN$, $FP$ and $FN$ refer to true positive, true negative, false positive and false negative, respectively. $TP$ and $TN$ represent the number of pixels which are classified correctly into crater rim contours and non-crater rim contours, respectively, in the prediction map. $FP$ represents the number of pixels that are predicted as crater rim but are labeled as non-crater rim in ground truth annotations. $FN$ represents the number of pixels that are actually labeled as crater rim but are mistakenly classified as non-crater rim contours.

For instance-level evaluations, we first need to match the predicted craters to the ground truth. To this end, we reuse Equation (6) to effectively match predictions with annotations. After matching, the number of $TP$, $TN$ and $FP$ craters can be calculated and used in the calculation of recall, precision and $F_1$-score, which are chosen as instance-level metrics for assessment of predicted craters by template matching. Recall is the ratio of correctly predicted Martian craters to all Martian craters in annotations and is defined by Equation (8):

$$Recall = \frac{TP}{TP + FN}. \tag{8}$$

Precision is the ratio of correctly predicted Martian craters to the total predicted Martian craters (see Equation (9)):

$$Precision = \frac{TP}{TP + FP}. \tag{9}$$

$F_1$-score takes both false positives and false negatives into account. It is the weighted average of the precision and recall and is defined by Equation (10):

$$F_1 - score = 2 \cdot \frac{Recall \cdot Precision}{Recall + Precision}. \tag{10}$$

### 3.3. Hyperparameter Setting

In our paper, the learning rate of MC-UNet is set to $1 \times 10^{-4}$, and the batch size is set to 10. We use Adam optimizer and ReLU as the activation function to train the proposed MC-UNet because of better convergence performance and fewer parameters for tuning. We use TensorFlow 1.3 and Keras 2.26 to set up the deep leaning framework. The proposed MC-UNet is trained and tested on a deep learning server with a 2.3 GHz Intel Xeon Gold 5218 CPU, 128GB main memory and an NVIDIA Tesla V100 PCI-E GPU with 32 GB graphics memory powered by CUDA 10.2.

### 3.4. Structural Experiments

Setting the appropriate depth and width can improve the segmentation performance if the network is tailored to the specific task. For instance, U-Net is designed for medical image segmentation and has four logical layers for the encoder and decoder. The convolutional

channels are increased significantly from 64 to 1024 while downsampling feature maps. Crater U-Net is tasked with Martian crater segmentation and has seven convolutional layers, and the convolutional channels are increased from 16 to 128 during downsampling. Because of different network structures, they show extremely distinct performance in terms of recognition rate, training efficiency and parameter size. More precisely, U-Net needs fewer than 10 epochs to converge because of its channel width advantage. However, the size of the created model's parameter is approximately 400 MB, which is 50 times larger than that generated by Crater U-Net. Crater U-Net needs at least 500 epochs to reach its highest validation accuracy because of the lightweight convolutions and heavily imposed dropout regulations.

To leverage the advantages of U-Net and Crater U-Net, we use U-Net as the backbone to construct six candidate models with depths ranging from 2 to 7, and these models' widths are raised from 16 to 128, which is the width typically adopted by Crater U-Net. For each candidate model, similar to U-Net, we retain convolutional operations twice within each layer, followed by batch normalization. To speed up convergence, we omit the dropout regulations adopted in Crater U-Net. The performances of these six candidate models with varying depths are evaluated quantitatively and qualitatively, as shown in Table 1 and Figure 6, respectively. By analyzing the performance results, the optimal depth can be inferred.

**Table 1.** The crater segmentation statistics for six candidate models with depth from 2 to 7.

| Model | Channel/Depth | Accuracy | Training Time (s) | Model Size (MB) | Epoch | Recall | Precision | $F_1$-Score |
|-------|---------------|----------|-------------------|-----------------|-------|--------|-----------|-------------|
| 1 | (16→64→128) / 2 | 0.9803 | 148 | 6.69 | 50 | 0.6426 | **0.9540** | 0.7679 |
| 2 | (16→32→64→128) / 3 | 0.9819 | 124 | 7.12 | 40 | 0.7060 | 0.9045 | 0.7930 |
| 3 | (16→32→48→64→128) / 4 | 0.9821 | 85 | 8.19 | 30 | **0.7157** | 0.8991 | 0.7970 |
| 4 | (16→32→48→64→96→128) / 5 | 0.9819 | 83 | 11.87 | 25 | 0.7055 | 0.9024 | 0.7919 |
| 5 | (16→32→48→64→72→96→128) / 6 | 0.9822 | 96 | 15.88 | 20 | 0.6967 | 0.9435 | **0.8015** |
| 6 | (16→32→48→64→72→96→128→128) / 7 | **0.9824** | 103 | 27.46 | 15 | 0.7038 | 0.8768 | 0.7808 |

It can be observed from Table 1 that the semantic accuracy is improved with the increase in depth. The highest semantic accuracy reaches 0.9824 at the maximum depth of 7. For instance-level segmentation of impact craters, the model candidates with depth from 3 to 7 have similar performance, achieving around 0.70 recall and about 0.79 $F_1$-score. This implies that the crater recognition rate cannot be easily improved by separately increasing the depth of the model. Despite this, an extremely shallow neural network is also not acceptable. For example, as shown in Table 1, the candidate model with depth of 2 has the worst recall of 0.6426 due to a lack of sufficient receptive field size, which is mainly determined by the depth of the network.

To further evaluate the performance of candidate models with different depths, we randomly select three small regions from the test tiles of THEMIS 2006 and qualitatively compare the characteristics of the recognized impact craters, as shown in Figure 6. It can be seen that model candidates with depths of 5 or 6 are more likely to recognize the medium- and large-sized impact craters compared with model candidates with depths of 4 and 7, as highlighted by red rectangles in Figure 6. This proves that the candidate models with depths of 5 and 6 are relatively more suitable for crater recognition on Mars.

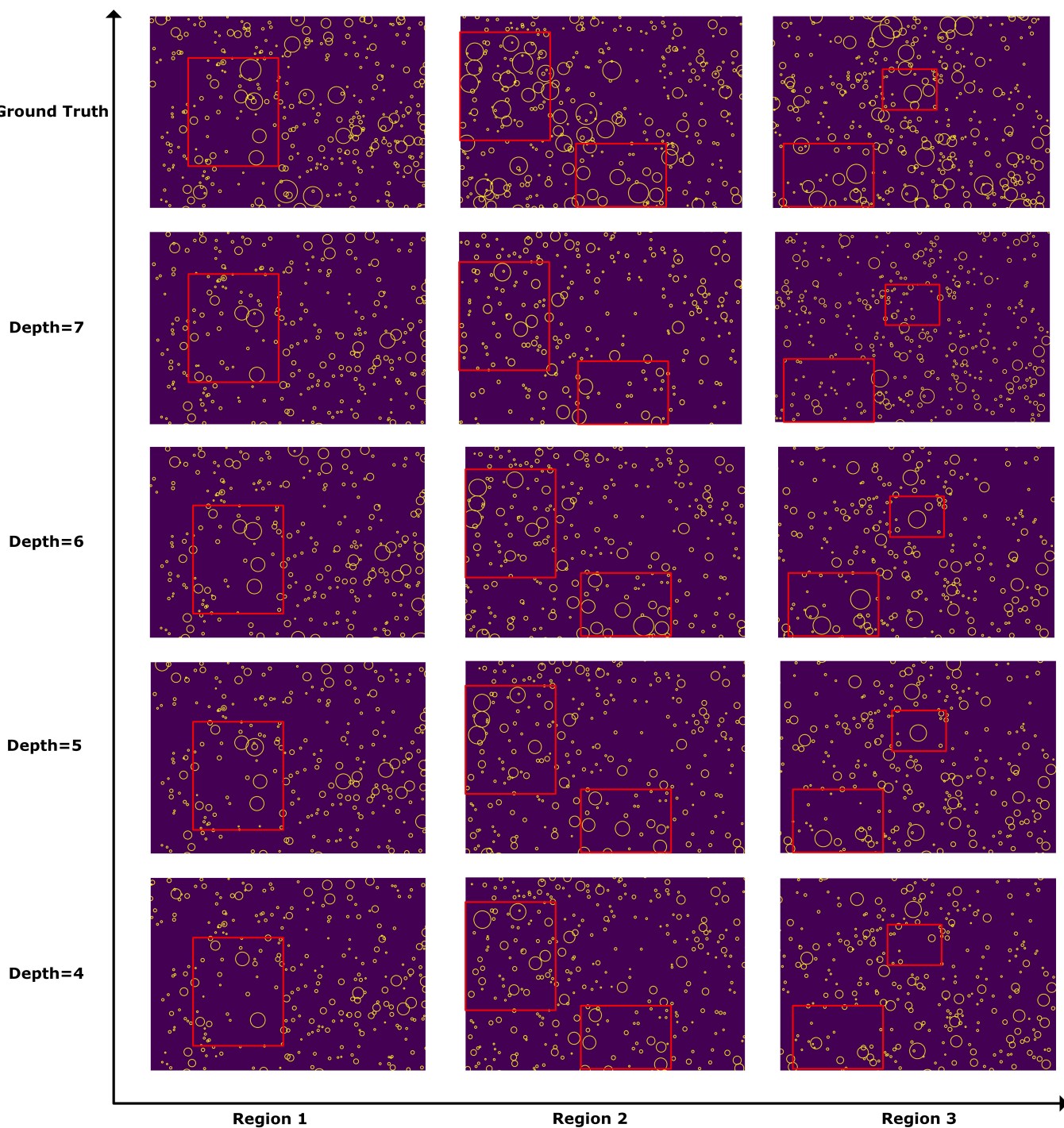

**Figure 6.** Qualitative performance comparisons with candidate models of different depths. Note that candidate models with depths of 5 and 6 tend to successfully recognize medium- and large-sized impact craters.

We further conduct width experiments to verify what range of channels is most beneficial for crater recognition on Mars. To this end, we fix the best two depths, i.e., depths of 5 and 6, and then expand the model's default width (16∼128) by multiplying three expansion coefficients $\omega$, e.g., 1, 1.5 and 2. The quantitative results are reported in Table 2, from which we can see that the semantic accuracy is improved with the expansion of model width. For any candidate models with depths of 5 or 6, the best two models correspond to an expansion coefficient of 2. That is, we obtain the highest semantic accuracies of 0.9832 and 0.9829 for models 3 and 6, respectively. Instance-level segmentation shows the same

trend. Model 3 with a depth of 5 and twice the width achieves the highest $F_1$-score of 0.8247. Model 6 with a depth of 6 and twice the width reaches the best recall of 0.7752. It can be expected that when the expansion coefficient surpasses a value of 2, we might obtain a higher recognition rate of impact craters. However, considering that the model will have an extremely large number of parameters and high training cost, it is inefficient to train large-scale input images. For the task of impact-crater recognition on Mars, it can be concluded that the model structure with depth of 5 and expansion coefficient of 2 constitutes a good trade-off between multiple factors such as accuracy, training efficiency and parameter size.

**Table 2.** Qualitative performance comparisons with candidate models of different widths. The symbols '#DC' and '#MC' represent the number of the detected craters and the number of successfully matched craters, respectively. The abbreviation 'EC' means expansion coefficient.

| Model | Channel/Depth | EC | Accuracy | Training Time (s) | Model Size (MB) | Epoch | Recall | Precision | $F_1$-Score | #DC | #MC |
|-------|---------------|-----|----------|-------------------|-----------------|-------|--------|-----------|-------------|------|------|
| 1 | (16→32→48→64→96→128) / 5 | 1.0 | 0.9824 | 83 | 11.87 | 25 | 0.7055 | 0.9024 | 0.7919 | 1763 | 1591 |
| 2 | (24→48→72→96→144→192) / 5 | 1.5 | 0.9827 | 124 | 30.83 | 20 | 0.6448 | 0.9642 | 0.7728 | 1508 | 1454 |
| 3 | (32→64→96→128→192→256) / 5 | 2.0 | **0.9832** | 136 | 54.49 | 20 | **0.7374** | 0.9353 | **0.8247** | 1778 | 1663 |
| 4 | (16→32→48→64→72→96→128) / 6 | 1.0 | 0.9822 | 96 | 15.88 | 20 | 0.6967 | 0.9435 | 0.8015 | 1665 | 1571 |
| 5 | (24→48→72→96→108→144→192) / 6 | 1.5 | 0.9822 | 142 | 38.22 | 20 | 0.7614 | 0.8725 | 0.8132 | 1968 | 1717 |
| 6 | (32→64→96→128 →144→192→256) / 6 | 2.0 | **0.9829** | 159 | 108.16 | 15 | **0.7752** | 0.8543 | 0.8128 | 2046 | 1748 |

*3.5. Mechanism Experiments*

When the structure of the network is determined, we further conduct mechanism experiments to infer which mechanisms do really contribute to impact-crater recognition on Mars. In this paper, we adopt three types of mechanisms, including downsampling strategy, encoder–decoder feature fusion style and attention mechanism to strengthen the performance of MC-UNet.

(1) Downsampling strategies

As previously mentioned in Section 2.3, we adopt three downsampling strategies, namely max pooling, average pooling and stem-block pooling (see Figure 7) to encode the semantic features. The quantitative performance statistics for these three pooling strategies are shown in Table 3. In terms of crater recognition, max pooling and average pooling yield approximately equivalent $F_1$-score accuracies of 0.8274 and 0.8184, respectively. Max pooling obtains the maximum $F_1$-score of 0.8247, while average pooling acquires the highest recall of 0.7796. Stem-block pooling has the lowest instance accuracy with an $F_1$-score of 0.8073, which is mainly because the convolution operations contained in the stem block module cannot play a significant role in compensating for the lost information. In fact, in our encoder–decoder architecture, the skip connection architecture has already integrated the detailed appearance information into upsampled feature maps. Therefore, the convolutional operations in the stem-block module appears to be redundant. In terms of computational efficiency, the training cost per epoch of stem blocks is far more than for the other two poolings because of externally introduced convolutional operations within the stem-block module.

**Table 3.** The quantitative performance statistics for three pooling strategies.

| Downsampling | Accuracy | Training Time (s) | Model Size (MB) | Epoch | Recall | Precision | $F_1$-Score |
|--------------|----------|-------------------|-----------------|-------|--------|-----------|-------------|
| Max Pooling | 0.9832 | 136 | 54.49 | 20 | 0.7374 | 0.9353 | **0.8247** |
| Average Pooling | **0.9835** | 130 | 54.49 | 20 | **0.7796** | 0.8613 | 0.8184 |
| Stem-Block Pooling | 0.9835 | **186** | 58.43 | **170** | 0.7211 | **0.9356** | 0.8073 |

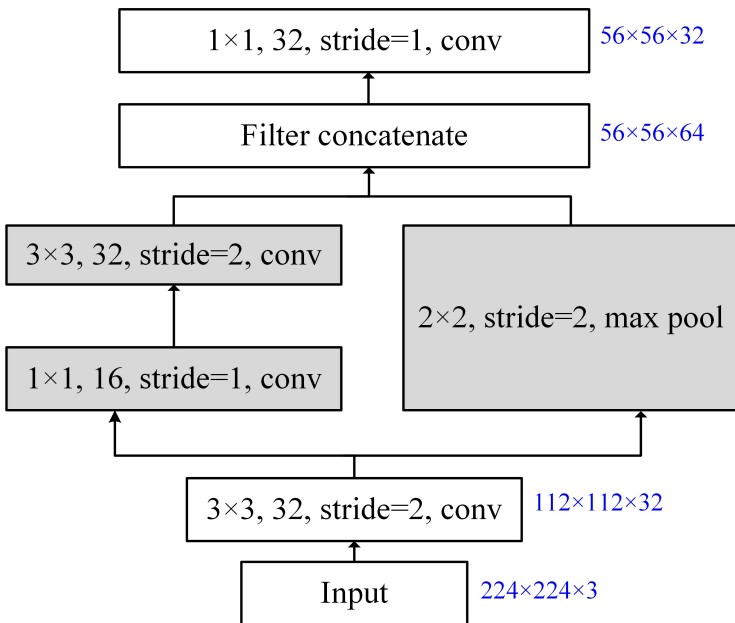

**Figure 7.** Basic structure of the stem-block module [38].

We further implement a qualitative comparison between max pooling and average pooling to exhibit their different performances for impact-crater recognition. As shown in Figure 8, it can be observed that max pooling tends to recognize large-sized impact craters, while average pooling is more sensitive to small-sized impact craters. This is probably because during backpropagation of the max pooling, the gradient from the next layer is passed back to only that neuron that achieves the max, and all other neurons get zero gradients. The neurons whose gradients are zero do not contribute to the gradients in the earlier neurons due to the chain rule. This simply means there are no gradients in positions without maximum values. Because of this, adopting max pooling is more likely to retain large-sized craters with prominent crater rims. However, for backpropagation of average pooling, more of the gradient from the next layer can pass back to all neurons in the earlier layer. This can help retain more active neurons and make the gradient signal backpropagate further. Because of this, some small-sized impact craters, which generally have weak rims, are more likely to be recognized, thereby reaching the highest recall of 0.7796, as shown in Table 3. As max pooling is sensitive to large craters, it results in higher instance-level precision of 0.9353.

In practice, during the task of crater recognition, we need to select the best pooling style according to different goals. For instance, age-dating of planetary surfaces requires counting the number of impact craters with varying sizes in some specific regions. In this case, average pooling, which has the potential to recognize as many impact craters as possible, is more appropriate. However, for analyzing interior and ejecta morphologies and for studying the degradation states of some representative craters, it is more meaningful to accurately trace large crater rims with complex features through max pooling.

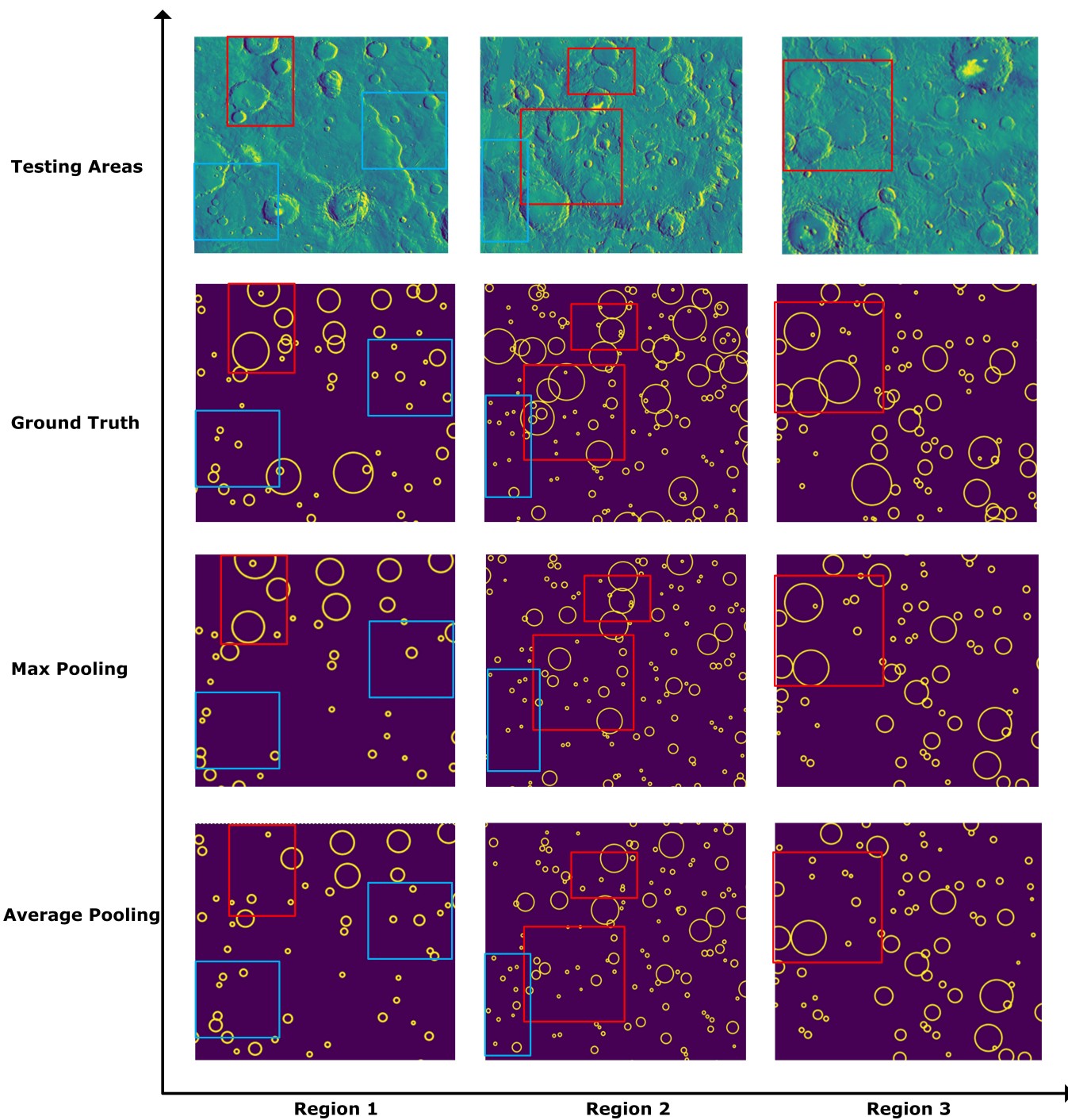

**Figure 8.** Craters recognized by max pooling and average pooling. It can be seen that max pooling is more likely to recognize large craters with complex features, while average pooling tends to find small craters with blurred rims.

(2) Feature fusion and attention mechanisms

We introduce two types of feature-fusion mechanisms, namely element-wise addition and concatenation, as well as three types of attention, namely channel attention, spatial attention and hybrid attention, into our gradually enriched U-Net structure model to analyze these mechanisms' performance on crater recognition. As shown in Table 4, it can be observed that concatenation fusion style is absolutely superior to element-wise addition in terms of semantic accuracy, instance accuracy and training efficiency. Therefore, we

adopt concatenation for feature fusion and then make further combinations with three other attentions to jointly evaluate their crater-detection performance.

**Table 4.** The quantitative performance statistics for two feature fusion styles and three attention mechanisms. The abbreviation 'Concat.' means concatenation fusion style.

| Model | Configuration | Accuracy | Training Time (s) | Model Size (MB) | Epoch | Recall | Precision | $F_1$-Score |
|---|---|---|---|---|---|---|---|---|
| 1 | Concat. | 0.9832 | 136 | 54.49 | 20 | 0.7374 | 0.9353 | 0.8246 |
| 2 | Add | 0.9830 | 187 | 47.53 | 20 | 0.7166 | 0.9363 | 0.8118 |
| 3 | CA+Concat. | 0.9833 | 240 | 56.14 | 20 | 0.7654 | 0.9008 | **0.8276** |
| 4 | SA+Concat. | 0.9828 | 252 | 56.14 | 20 | 0.7162 | **0.9411** | 0.8134 |
| 5 | CBAM+Concat. | 0.9836 | 217 | 56.21 | 15 | **0.7752** | 0.8806 | 0.8245 |

As can be seen from Table 4, the recognition accuracy of Model 3 achieves the best performance of 0.8276 in terms of $F_1$-score, which implies that the introduced channel attention exhibits its effectiveness. However, it should be noted that the imposed spatial attention in Model 4 does not improve the crater recognition rate although it acquires the highest precision of 0.9411.

We also qualitatively compare the predicted results of three arbitrarily selected regions from the test tiles by combining concatenation with three other attentions. As can be seen from Figure 9, channel and hybrid attentions can better help the model identify large craters, as demonstrated by the red rectangles. However, Model 4 embedded with spatial attention does not exhibit obvious advantages in either large Martian craters with complex features or small ones, which might have fuzzy and broken rims. This is probably because the crater distribution in the selected regions is relatively even, which extremely weakens the importance of the regions of interest of craters. Considering that Model 3 with channel attention achieves the highest $F_1$-score of 0.8276 and has better potential in recognizing large craters, we employ channel attention and concatenation feature fusion mechanisms into the proposed MC-UNet.

### 3.6. Effects of Structure–Mechanism Combinations

In Sections 3.4 and 3.5 above, we only evaluate the structure or the mechanism itself and ignore their combination effects. In this section, the previously evaluated promising structures and mechanisms are combined, from which the best configuration is determined to be the final framework of MC-UNet. More precisely, we select two depths, i.e., depth 5 and depth 6, as well as one width with an expansion coefficient of 2 from the perspective of the network's structure. In addition, we choose two types of downsampling strategy, i.e., max pooling and average pooling, two kinds of attention mechanism, i.e., channel attention and hybrid attention, and a concatenation fusion style from the perspective of the network's mechanism.

The predicted Martian crater results derived from the 12 typical configurations/models are shown in Table 5. On this basis, we show the comparative histogram of Martian crater accuracy in descending order of recall, which determines the match ratio to a great extent, as can be seen in Figure 10. Although Model 4 achieves the highest recall of 0.7796 by recognizing 2041 Martian craters for the test tile, it should be noted that 283 of the 2041 carters are mistakenly classified, leading to low precision of 0.8613. In this case, the model needs to spend too much time identifying these pseudo craters from the predictions. In contrast, Model 10 has the highest precision of 0.9449; however, it has an extremely low recognition rate (recall) of 0.7220 due to missing a lot of craters. $F_1$-score is also considered as a valuable metric to measure recognition performance by balancing recall and precision. In this sense, Models 11, 6 and 5 take the first three places in descending order of $F_1$-score, but in practice, the first two models are purposely excluded with due consideration of recall, lightweight structure and computational efficiency. That is, the selected Model 5 can guarantee prediction accuracy while having a compact structure and high computational

performance. Further, the channel attention adopted in Model 5 enhances the model's capability to recognize large craters, and average pooling tends to be sensitive to small craters' rims. Lightweight and efficient structures and effective mechanisms used in Model 5 jointly guarantee the best performance. Therefore, Model 5 is chosen as the final framework of MC-UNet.

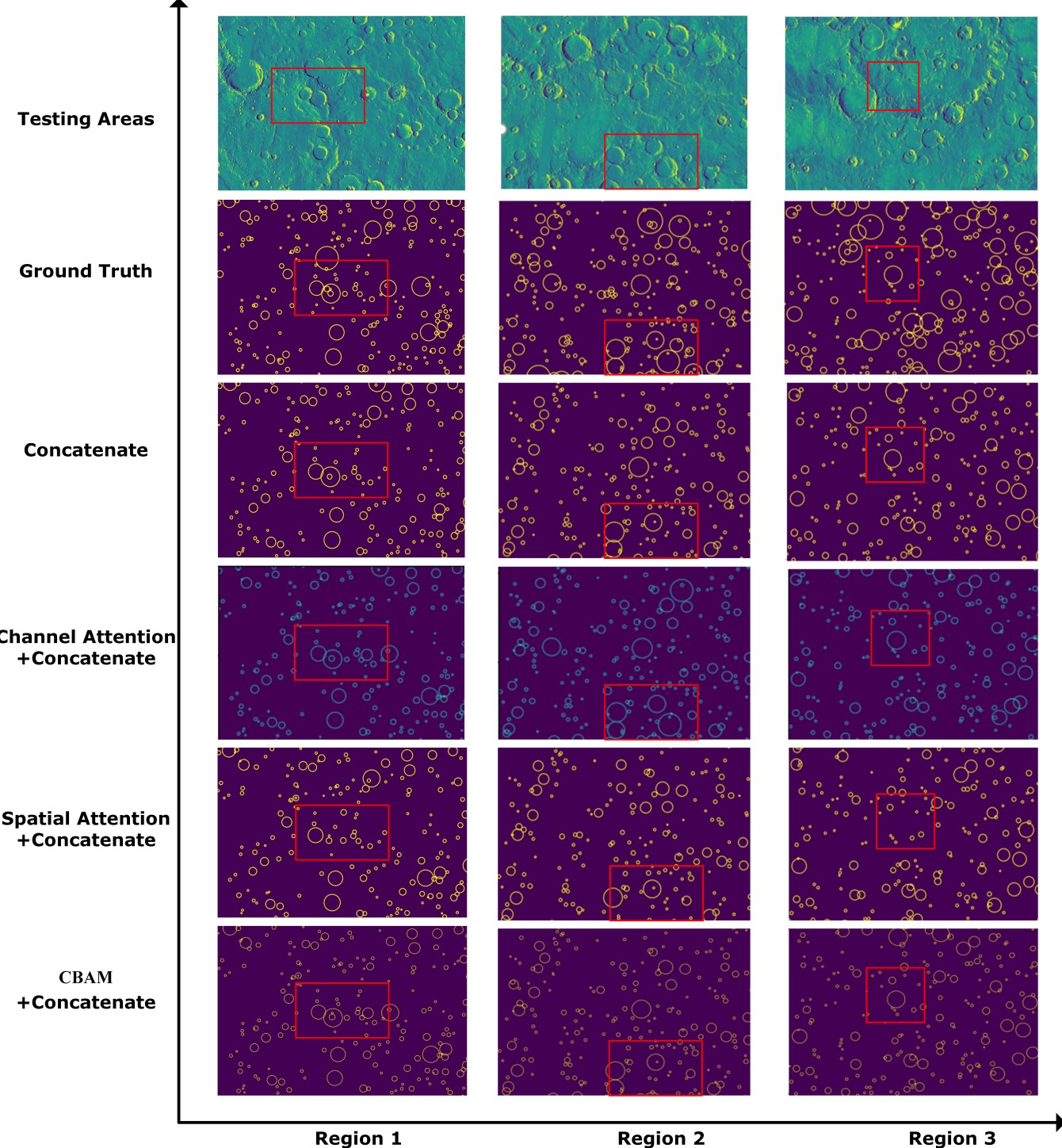

**Figure 9.** The predicted Martian crater results by combining concatenation feature fusion with three other attention mechanisms. It can be seen that channel and hybrid attentions tend to recognize the large Martian craters highlighted by the red rectangles.

**Table 5.** The combination of promising structures and mechanisms. The symbols '#DC' and '#MC' represent the number of the detected craters and the number of successfully matched craters, respectively. Other symbols 'MP', 'AP', 'CA', 'SA', 'CBAM' denote max pooling, average pooling, channel attention, spatial attention and CBAM hybrid attention, respectively. 'Concat.' means concatenation fusion.

| Model | Configuration | Recall | Precision | $F_1$-Score | #DC | #MC |
|---|---|---|---|---|---|---|
| 1 | 5/MP/Concat. | 0.7374 | 0.9353 | 0.8246 | 1778 | 1663 |
| 2 | 5/MP/Concat./CA | 0.7654 | 0.9008 | 0.8276 | 1916 | 1726 |
| 3 | 5/MP/Concat./CBAM | 0.7752 | 0.8806 | 0.8245 | 1985 | 1748 |
| 4 | 5/AP/Concat. | **0.7796** | 0.8613 | 0.8184 | 2041 | 1758 |
| 5 | 5/AP/Concat./CA | 0.7761 | 0.8910 | 0.8296 | 1964 | 1750 |
| 6 | 5/AP/Concat./CBAM | 0.7610 | 0.9181 | 0.8322 | 1869 | 1716 |
| 7 | 6/MP/Concat. | 0.7752 | 0.8543 | 0.8128 | 2046 | 1748 |
| 8 | 6/MP/Concat./CA | 0.7304 | 0.9120 | 0.8111 | 1806 | 1647 |
| 9 | 6/MP/Concat./CBAM | 0.7631 | 0.9029 | 0.8272 | 1906 | 1721 |
| 10 | 6/AP/Concat. | 0.7220 | **0.9449** | 0.8185 | 1723 | 1628 |
| 11 | 6/AP/Concat./CA | 0.7716 | 0.9034 | **0.8323** | 1926 | 1740 |
| 12 | 6/AP/Concat./CBAM | 0.7614 | 0.8957 | 0.8231 | 1917 | 1717 |

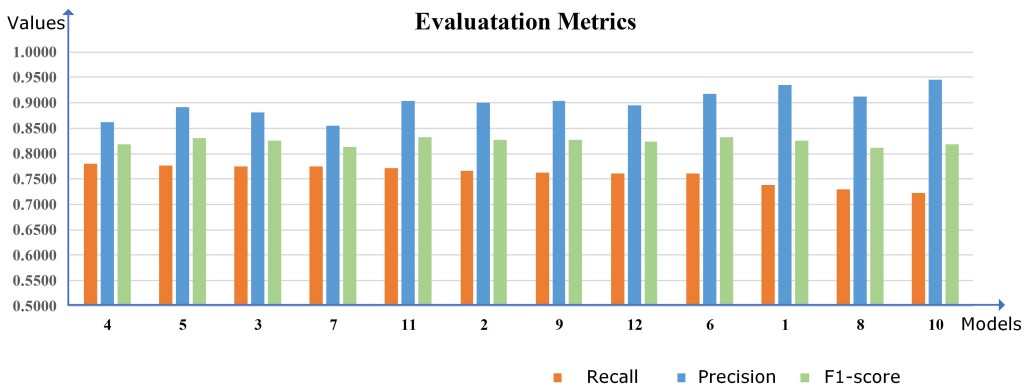

**Figure 10.** Comparative histograms of Martian crater accuracy in descending order of recall.

*3.7. The Optimal Number of Training Epochs*

Once the framework of MC-Net has been determined, the next step is to tune the number of training epochs to strike a trade-off between recognition accuracy and time efficiency. To this end, we use a batch size of 10 as well as epochs of 20, 50, 100 and 150 to train MC-UNet four times to determine the optimal number of epochs. The quantitative comparison results at semantic and instance levels are shown in Table 6, from which we can clearly see that the the best instance-level performance occurs at 100 epochs. At that point, MC-UNet obtains the highest recall of 0.7916 and the maximum $F_1$-score of 0.8355. If we continue to increase the number of training epochs, we observe a drop in accuracy. This possibly implies that the model achieves the best prediction performance at around 100 epochs, and continuously increasing the number of epochs might cause overfitting.

In addition, it should be noted that MC-UNet trained with different numbers of epochs exhibits different capabilities for recognizing large-sized craters. The detected maximum radii are recorded in Table 6 in the rightmost column, from which we can observe that the maximum radius of 136 pixels is recognized at 100 training epochs. This means that MC-UNet trained at 100 epochs not only has high recognition rate but also has the capability to perceive the large-sized craters as much as possible.

As shown in Table 6, we also compare the proposed MC-UNet trained at 100 epochs with Crater U-Net trained at 500 epochs. It can be seen that they have approximately equal values of recall. The correctly recognized craters by Crater U-Net are only four more than those predicted by MC-UNet. However, MC-UNet outperforms Crater U-Net in the aspect of precision, $F_1$-score and MR. Figure 11 shows the qualitative comparisons between

MC-UNet and Crater U-Net. It can be seen clearly that MC-UNet trained at 20 epochs can recognize only one crater from the two representative Martian craters indicated by two red rectangles in Figure 11a. When the number of epochs reaches 100, our model can perceive these two large craters, as demonstrated in Figure 11b. However, none of them can be successfully recognized by Crater U-Net although it is trained up to 500 epochs, as evident in Figure 11c. To make the predicted circles clearer, we use different colors in Figure 12 to label true positive (TP), false positive (FP) and false negative (FN) circles. It can be observed that most of the crater-omission errors denoted by green occur at craters with obviously fuzzy rims, while the crater-ommission errors denoted in yellow appear to be extremely small.

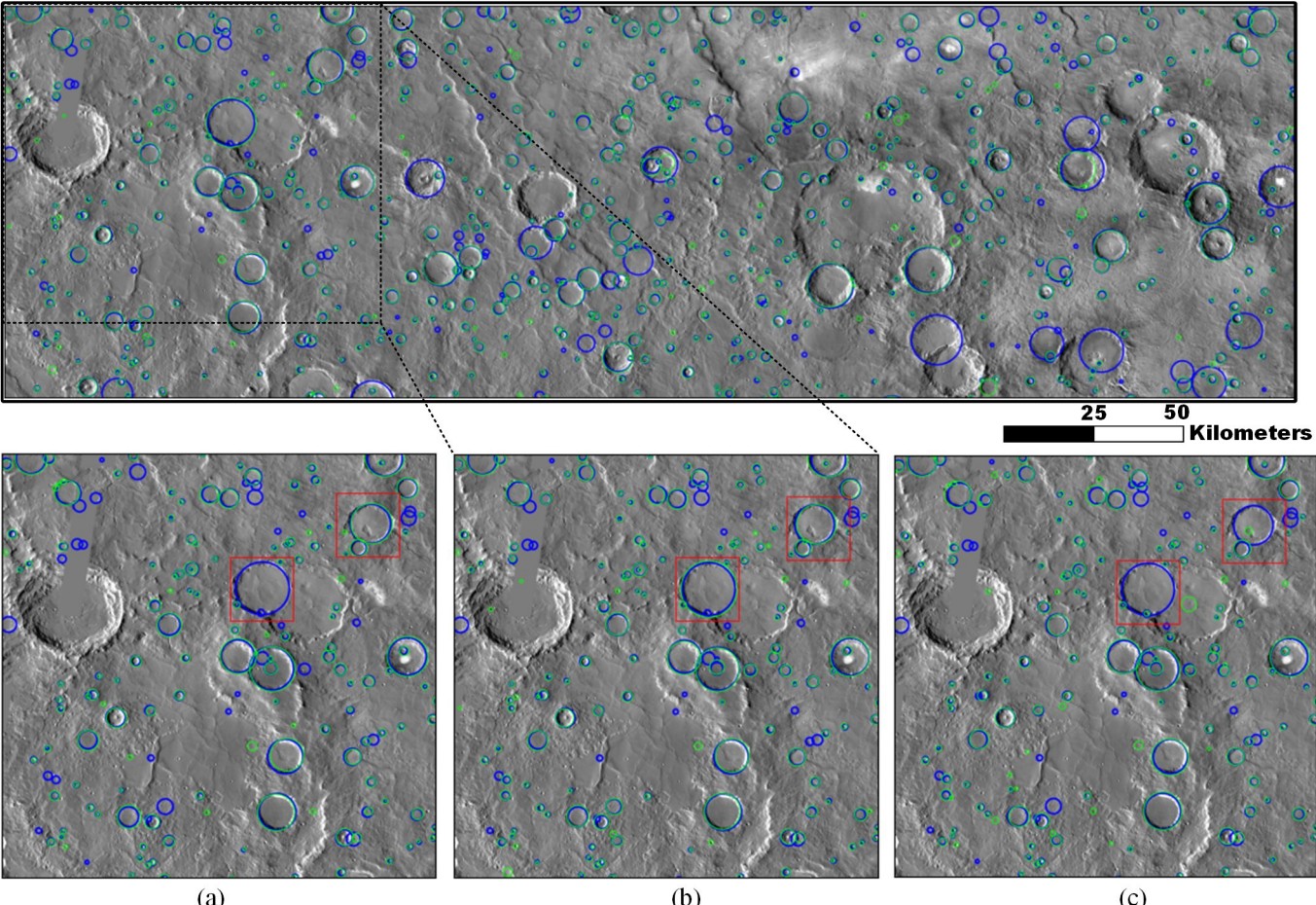

(a)
(b)
(c)

**Figure 11.** The performance comparisons between MC-UNet and Crater-UNet trained at different numbers of epochs. The predicted craters are indicated in green, while the ground truth annotations are labeled in blue. (**a**) MC-UNet prediction results with 20 training epochs. (**b**) MC-UNet prediction results with 100 training epochs. (**c**) Recognized craters by Crater U-Net with training epochs at 500. Note that for two typical larger craters denoted by two red rectangles, MC-UNet with 20 epochs can only recognize one of them; MC-UNet with 100 epochs can successfully recognize both of them; however, Crater U-Net cannot recognize either of them.

**Table 6.** Quantitative comparison of MC-UNet and Crater U-Net based on different numbers of training epochs. The symbols '#DC' and '#MC' represent the number of the detected craters and the number of successfully matched craters, respectively. 'MR' denotes the maximum radius of the craters.

| Model | Epoch | Accuracy | Recall | Precision | $F_1$-Score | #DC | #MC | MR (pixels) |
|---|---|---|---|---|---|---|---|---|
| | 20 | 0.9835 | 0.7761 | 0.8910 | 0.8269 | 1964 | 1750 | 123 |
| | 50 | 0.9830 | 0.7880 | 0.8780 | 0.8306 | 2024 | 1777 | 125 |
| MC-UNet | 100 | 0.9833 | **0.7916** | 0.8845 | **0.8355** | 2018 | 1785 | **136** |
| | 150 | **0.9836** | 0.7698 | **0.8935** | 0.8271 | 1943 | 1736 | 126 |
| Crater U-Net | 500 | 0.9820 | 0.7933 | 0.8787 | 0.8338 | 2036 | 1789 | 128 |

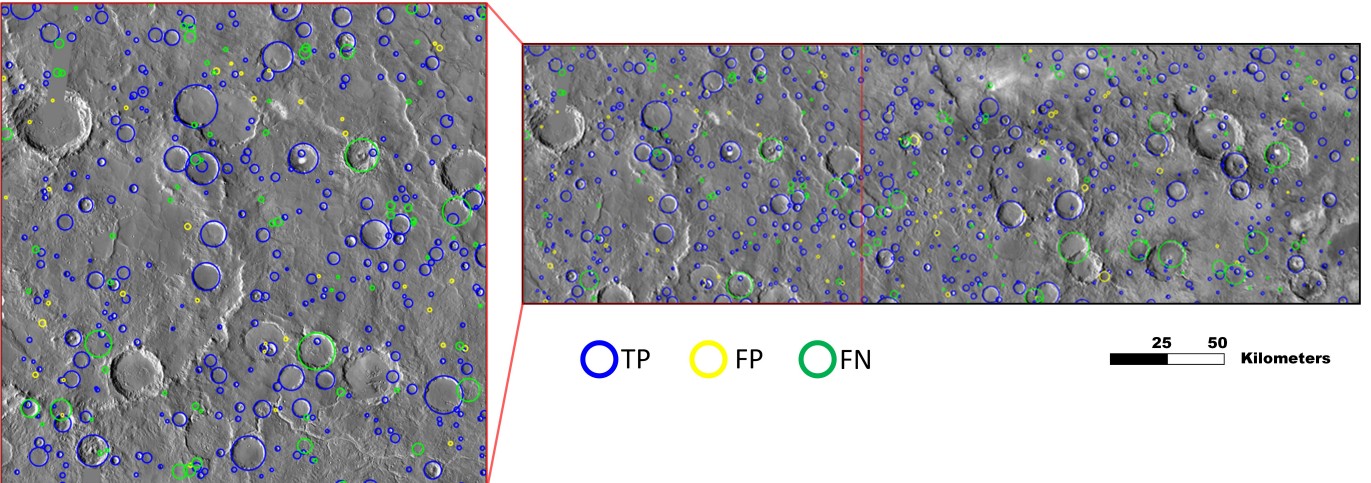

**Figure 12.** The predicted craters by the proposed MC-UNet trained at 500 epochs. Note that the blue, green and yellow circles represent true positive, false negative and false positive results.

### 3.8. Performance Comparison with State-of-the-Art Methods

We make further comparisons with other state-of-the-art classic networks, such as U-Net and Crater U-Net. The qualitative and quantitative comparisons are shown in Table 7 and Figure 13, respectively. From Table 7, it can be seen that MC-UNet and U-Net are obviously superior to Crater U-Net in the aspect of semantic accuracy. For instance-level performance, even though U-Net achieves the maximum precision of 0.9183, MC-UNet and Crater U-Net have an absolute advantage over U-Net in recall. In practice, in the task of crater counting, the metric recall is considered to be the most important and effective measure because to a great extent it represents the match ratio, the percentage of predicted craters in annotations. Further, the models have significant differences in computational efficiency. For example, U-Net has an extremely large number of parameters because the the maximum convolutional channel (width) of U-Net reaches up to 1024, thereby requiring 393 s training time per epoch. Crater U-Net is lightweight with only a few parameters. Therefore, it just needs 30 s training cost per epoch but requires 500 epochs to get the best performance. In contrast, the proposed MC-UNet needs only 20 epochs to achieve convergence and at most 100 epochs to obtain the best predicted performance, thereby indicating a faster convergence rate.

We further analyze the performance of these three models for recognizing large-sized Martian craters. As shown in Figure 13, it can be observed that U-Net scarcely recognizes large impact craters. Crater U-Net can recognize relatively large-sized craters reluctantly; however, the proposed MC-UNet can almost completely identify these large-sized craters. From Table 7, we can see that in our test tile, U-Net can predict craters with a maximum radius of 124 pixels, and the maximum is 128 pixels for Crater U-Net. However, our model can successfully identify craters with a maximum radius of 136 pixels.

**Table 7.** Quantitative comparison of three convolutional neural networks using the test tile from THEMIS 2006 in Figure 5. The symbol 'MR' denotes the maximum radius of the craters. *RMSE* means the crater location accuracy between the prediction and the ground truth. We only use matched crater pairs for calculation of *RMSE*.

| Model | Accuracy | Training Time (s) | Model Size (MB) | Epoch | Recall | Precision | $F_1$-Score | MR (pixels) | *RMSE* (m) |
|---|---|---|---|---|---|---|---|---|---|
| U-Net | **0.9838** | 393 | 395.46 | 25 | 0.7681 | **0.9183** | 0.8365 | 124 | 55.02 |
| Crater U-Net | 0.9820 | 30 | 8.51 | 500 | **0.7933** | 0.8787 | 0.8338 | 128 | 54.53 |
| MC-UNet | 0.9833 | 156 | 56.14 | 100 | 0.7916 | 0.8845 | 0.8355 | **136** | 54.67 |

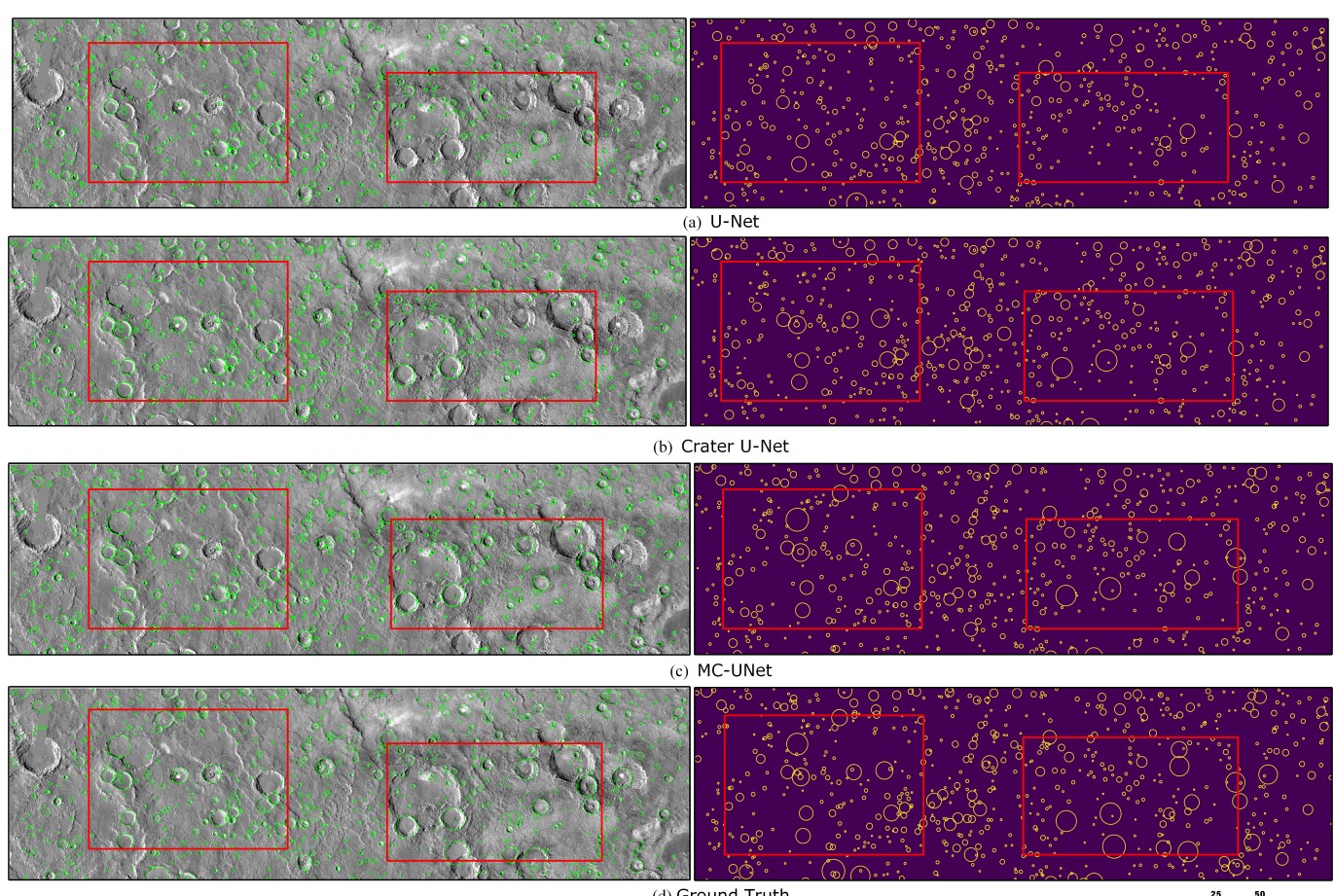

**Figure 13.** The performance comparisons between U-Net, Crater U-Net and MC-UNet. The denoted red rectangles represent the regions where the large-sized craters are gathered and can be easily detected by MC-UNet and Crater U-Net. The craters from (**a**–**c**) are predicted by U-Net, Crater U-Net and MC-UNet, respectively, and (**d**) is the ground truth. It should be noted that the relatively large craters indicated by red rectangles are easily found by MC-UNet and Crater U-Net.

### 3.9. Performance Comparison on Two Time Periods of Datasets

To verify the robustness of the proposed MC-UNet, we compare and analyze the prediction performance using two time series of THEMIS infrared images that were released in 2006 and 2010. The resolution and dimensions of each divided tile are discussed in Section 3.1. If we directly crop the THEMIS 2010 images into patches with a size of $512 \times 512$ pixels as input, the medium- and large-sized craters might extend beyond the scope of the receptive field, thereby weakening the recognition ability of these remarkable craters by the proposed MC-UNet. To solve this problem, THEMIS 2010 infrared images are downsampled to the same resolution as THEMIS 2006. After downsampling, the test

tile is resized from the original 17,783 × 17,783 pixels to 7680 × 7680 pixels via bilinear interpolation. After that, we divide the test tile into 15 × 15 patches, each of which has dimensions of 512 × 512 pixels and is provided as input to MC-UNet.

The predicted results of these two time periods of THEMIS infrared images are shown in Figure 14. It can be seen that the shadow effects in THEMIS 2006 infrared images in Figure 14a,c are more prominent compared to THEMIS 2010 infrared images in Figure 14b. This is mainly caused by different illumination and incidence angles. In practice, shadow effects can help make the crater rims more prominent. Because of this, they are regarded as very useful clues to discover impact craters. Despite having weak shadows in THEMIS 2010, the proposed MC-UNet can successfully recognize most of the craters, achieving a recall of 0.7524 and $F_1$-score of 0.8114. Compared to the THEMIS 2006 dataset, the THEMIS 2010 daytime infrared mosaic image has fewer missing values due to adoption of an enhanced image-stitching technique. This enhancement makes the craters distributed in missing data regions be recognized successfully, as is proven in the enlarged views in Figure 14.

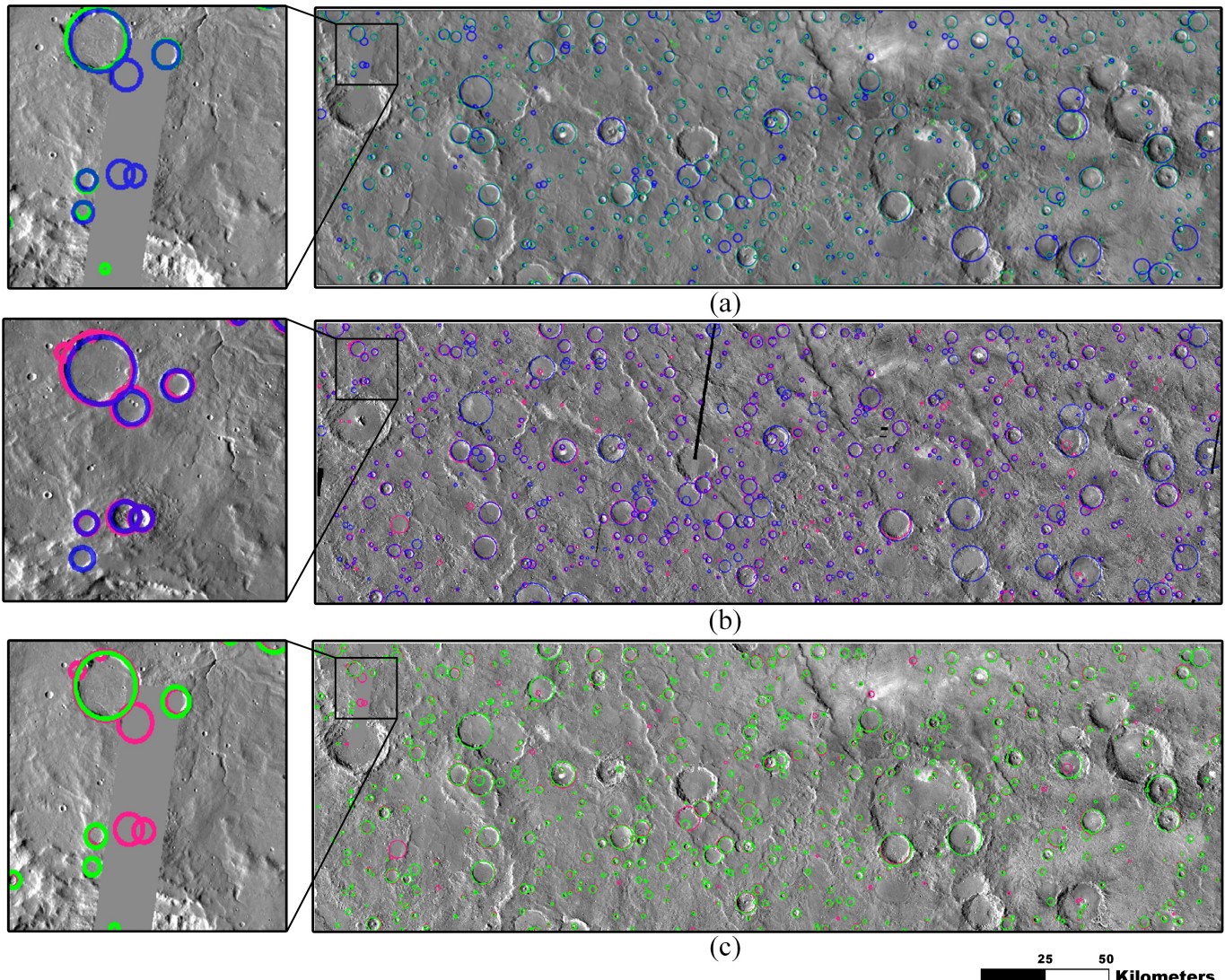

**Figure 14.** The comparison results of THEMIS daytime infrared images released in 2006 and 2010. Green and pink circles denote predicted craters from THEMIS 2006 and 2010 images, while blue circles represent the annotated craters. In subfigures (**a**,**b**), the predicted and ground truth craters from THEMIS 2006 and 2010 are superimposed onto their corresponding images. (**c**) The predicted craters from the THEMIS 2006 and 2010 images are overlaid onto the background of the THEMIS 2006 image.

## 4. Discussion

MC-UNet has six depth layers to provide a large enough receptive field for recognition of large-sized, complex Martian craters. Meanwhile, the maximum number of channels is reduced to one-fourth of U-Net, which effectively reduces the model's parameters and improves the training efficiency. In terms of mechanism, the proposed MC-UNet adopts the average pooling strategy, which tends to identify relatively small-sized Martian craters. In addition, feature fusion and channel attention mechanisms are embedded into the skip connections, which makes MC-UNet more sensitive to the rims of Martian craters.

Although the proposed MC-UNet is lightweight, highly efficient and accurate, there are some drawbacks that should be considered. The proposed MC-UNet cannot successfully recognize a wide range of variations (see Figure 15), such as extremely large-sized craters up to 100 km in diameter, craters with fuzzy or broken rims, partially overlapping craters, concentric craters, and other impact craters with complex features. To accurately recognize the large-sized craters in Figure 15a, in future work, we plan to combine machine learning and deep learning to hierarchically detect impact craters with varying sizes. That is, we plan to first employ machine learning to recognize the extremely large craters, and use MC-UNet to perceive other medium and most small craters. In this way, we can make our model recognize craters that are sized beyond the input size of 512 × 512. For the complex craters depicted in Figure 15b–d, we plan to combine THEMIS infrared images with the Mars' DEM dataset acquired from the Mars Orbiter Laser Altimeter (MOLA) [41] as MC-UNet's inputs to expand the input channel dimensions and improve complementarity between 2D and 3D data sources.

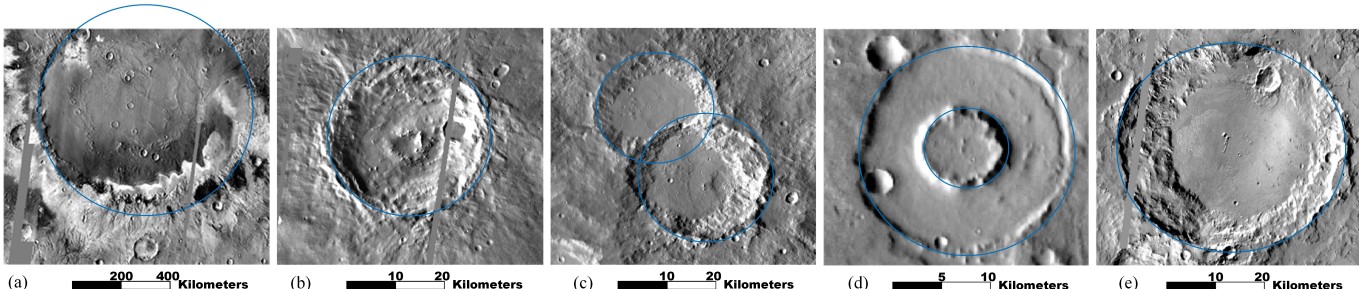

**Figure 15.** Some complex Martian craters that cannot be predicted by the proposed MC-UNet. In The complex impact craters denoted by blue circles from the leftmost to rightmost subfigures are (**a**) extremely large crater, (**b**) fuzzy-rim crater, (**c**) partially overlapping craters, (**d**) concentric craters and (**e**) terraced crater.

In addition, it should be noted that in our paper, impact craters are always fitted by a series of 2D circular templates, but in practice, some elongated craters may exist on Mars. In this cause, the fitting accuracy is not to be guaranteed by the inappropriate circular kernel. Therefore, we plan to put more kernel functions such as an elliptically shaped kernel into our template library to more accurately trace elliptically shaped crater rims. For other abnormal shapes of craters that cannot be represented by any kernel shape functions, we will use the concept of "hybrid representation" to fit regular and atypically shaped craters. That is, Martian craters will be accurately modeled by kernels for the regular shapes and by contour tracing for the atypical shapes.

To further verify the extensibility of the proposed network, in future work, we will apply our model to explore impact craters over the lunar surface and other terrestrial planets. We also have plans to investigate their morphological characteristics to obtain a better understanding of the formation process of these craters on different terrestrial planets.

## 5. Conclusions

In this paper, we propose a novel convolutional neural network called MC-UNet with U-Net as the backbone for recognition of impact craters on Mars at semantic and instance

levels. MC-UNet has a depth of six, and the max number of convolutional channels of 256, thereby keeping the entire framework lightweight and striking a good balance between prediction accuracy and training efficiency. In addition, some additional mechanisms, such as an attentive mechanism and feature-map fusion mechanism are embedded into MC-UNet by means of skip connection architecture to highlight large-sized impact craters with complicated features and to enhance the overall recognition rate of impact craters of various sizes.

The result shows that the proposed MC-UNet achieves a high recall rate of 0.7916 and precision of 0.8845, which commonly determine the high $F_1$-score of 0.8355. In addition, our MC-UNet can recognize a maximum radius of 31.28 km (136 pixels) craters from the test tile, which is far bigger than the largest craters predicted by classic U-Net.

**Author Contributions:** D.C. and F.H. analyzed the data and wrote C++ source code. Z.Z. and J.P. helped with project and study design, paper writing and data analysis. P.T.M. helped with proofreading. All authors have read and agreed to the published version of the manuscript.

**Funding:** This work was supported in part by the National Natural Science Foundation of China under grants 42271450, 41971415 and 42071445; in part by the Natural Science Foundation of Jiangsu Province under grant BK20201387; in part by the Qinglan Project of Jiangsu Province, China; and in part by the Key Laboratory of Land Satellite Remote-Sensing Applications, Ministry of Natural Resources of the People's Republic of China, under grant KLSMNR-G202209.

**Data Availability Statement:** Martian THEMIS image processing and mosaicking was completed by the Mars Space Flight Facility at Arizona State University. The processed datasets are released to the public via the link http://www.mars.asu.edu/data/.

**Conflicts of Interest:** The authors declare no conflict of interest.

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
