# Peer review of "MC-UNet: Martian Crater Segmentation at Semantic and Instance Levels Using U-Net-Based Convolutional Neural Network"

_remotesensing, doi:10.3390/rs15010266_

Round 1

Reviewer 1 Report

The authors of the manuscript propose a methodology for Martian crater detection based on THEMIS infrared images using UNet. It is an interesting topic. The presentation of the method is complete and the experiments are convincing. 

However, there are some aspects that need minor adjustments or clarifications:

1. Some of the figures among Figs. 7, 8, 9, 12 should consider using circles with different colors to represent TP, FP, TN, FN. Now it's difficult to compare and tell the differences.

2. The complex craters that  can not be predicted should be annotated in Fig. 14. 

3. The metrics used in the manuscript not reflect location precision. Is there better way to evaluate location accuracy? Or how the proposed approach would tackle such problem?

4. Where is your discussion part? You should clear discuss your results, discussion similar works; advantages and what can be improved, etc.

Author Response

Kindly see the attached file.

Reviewer 2 Report

Thanks for this nice article. The article is well-structured, and overall very clear. The suggestions below should fix the little gaps and minor mistakes I found.

There were some typos (eg, in caption of Fig.11: "NOT that for our...", "NOTE" would be right), but nothing that would mislead the reader. Getting completely rid of them is up to you.

Major issues

Figure captions: the explanation of figures content -- the colors used and their meaning (ground truth, detected, etc) -- is coming at the end of the caption' text; After you explain the results the figure is trying to show. This is wrong. You should first explain to the reader what he/she is seeing (eg, in Fig.13: "Green and pink circles denote predicted craters ..., while blue circles the annotated craters."), then you tell the reader what is going on in the figure(s) ("In (a) and (b) you can see.....").
This is happening in all figures, and this is not a matter of style. Please, fix that.

Predictions x Annotations matching: In line 403 you start talking about "instance-level" evaluation metrics (by the way, there is typo there: dash "-" is missing in "instance-level"), it is not clear though how do you match the ground truth to the predicted craters. Previously in the text, you have talked about "redundant Martian craters" and the centroid-distances matching process, and that was clear. But I didn't see you explaining how effectively you match predictions with annotations. After this point is explained, you can go to the metrics.

Template matching: In line 342, it is not clear what is the meaning of “candidates”; If “candidates” is the detected craters (not necessarily known previously) how could you say “until all (…) candidates (…) are recognized”? Or, maybe, you have to define better what “recognized” mean, because I see a potential infinite loop. Please, explain this point properly.

Equation 5: I understand from the expression that “\mu” is a function of (x,y), is that correct? If "yes", make that clear in the equation (left-side); If "no", then explain better in the text what is the format of “\mu”.

Minor issues

Figure 3: I think Fig.3 should go to section 3.5 where this framework is going to be explained. Here, as far as I could see, it is of no use.

Line 234: You say: “It is extremely difficult to significantly improve the craters segmentation accuracy by merely tuning the structure…”. My question is "Why (is it so difficult)?" What do you mean by “significantly”? Or, maybe you want to pose it differently: is it difficult, or is there a limit? This is a minor issue, but since you proposed the discussion, I would like to feed me with more information on that; I'm curious.

Line 193: You say “depth of a network refers to the times of downsampling”. This is a strange sentence. Shouldn’t it be “the NUMBER of times downsampling is performed” (or something like that)?

Line 188: “In a convolutional neural network, the structure roughly contain the network's depth and width...”, something is missing here. Or you may want to rephrase this to, eg, "Typically, the structure of a CNN is roughly described in terms of its depth and width..." (if I understood it properly). Please, adjust that.

2.1, MC-UNet framework: the first paragraph of that section sounds like conclusion. It is misplaced or should be reduced. You should talk about methodoloy but you are already giving the results. 

Introduction: You say "evolution of celestial bodies, rotation axes changes, magmatic field activities, and past catastrophic events" and "we can infer the relative geological age between different surface regions and analyze the change process of the planetary surface", do you have references for those uses? This is a minor request, I think those are interesting to refer to for non planetary/geologists; If you have those/some references, I would like to see them there.

Author Response

Kindly see the attached file.

Reviewer 3 Report

   Dear Authors

   I found your paper of great interest. You clearly explain the novelty of your work, and I have not doubts about it. Your paper deserves to be published to explain your new approach.

   In any case, I think that your paper can increase in interest for a general audience if you link in the introduction your ability to determine crater ages with some hot research nowadays, like e.g. old Martian terrains from which excavated craters can deliver to Earth ancient meteorites with evidence of aqueous alteration like e.g. ALHA84001 (see e.g. Moyano-Cambero et al., 2017), or the current evidence of fresh craters that deliver current day meteorites (Lagain et al., 2021). I think you could make a short point to easily implement the detection and identification of small, bright craters that have the potential to deliver current day meteorites to Earth from relatively young regions sampled by the Shergottites. 

Congratulations for your work and approach.

Suggested references:

Lagain A. et al. (2021) The Tharsis mantle source of depleted shergottites revealed by 90 million impact craters. Nature Comm., https://doi.org/10.1038/s41467-021-26648-3

Moyano-Cambero, C.E., et al. (2017) Petrographic and geochemical evidence for multiphase formation of carbonates in the Martian orthopyroxenite Allan Hills 84001, Meteoritics & Planetary Science 52, 1030-1047.

Author Response

Kindly see the attached file.
